# Overlooked riverine contributions of dissolved neodymium and hafnium to the Amazon estuary and oceans

Antao Xu [1] ✉, Ed Hathorne[1], Georgi Laukert [1,2,3] & Martin Frank [1]

The Amazon River accounts for 20% of global freshwater runoff and supplies vital trace metals to the Atlantic Ocean. Suspended particles within its plume are thought to partially dissolve, constituting a large potential source of metals, which is, however, not well constrained. Here we used combined neodymium (Nd) and hafnium (Hf) isotopes to disprove the release of Nd and Hf from particles as the cause of the observed dissolved concentration increases and isotopic variability across the plume. Instead, the changes reflect admixture of nearby Pará River freshwater with exceptionally high dissolved Nd and Hf concentrations contributing 45–100% of the riverine fraction to the southern and outer estuary. This result led us to develop an empirical relationship between riverine Nd concentration and pH to revise the global dissolved riverine Nd flux, which accordingly is at least three times higher than commonly used estimates. Future work should focus on contributions of low-pH rivers to global metal fluxes.

Vast amounts of nutrients, dissolved organic matter and trace metals are introduced into the Atlantic Ocean via the Amazon estuary, thereby increasing micronutrient levels[1] and enhancing productivity within the freshwater plume[2,3]. In the Atlantic Ocean, these micronutrients allow diazotrophs and phytoplankton to flourish and sequester large amounts of atmospheric $CO_2$[4,5]. The factors controlling the trace metal contents of the plume are thus crucial to constrain and include dissolved riverine inputs[6,7], estuarine processes (e.g., removal and addition)[8], and the exchange between particulate and dissolved phases[9]. Despite the high dissolved riverine concentrations of trace elements and the large volume of freshwater discharged by the Amazon River (~$6.6 \times 10^{12}$ $m^3$ $yr^{-1}$)[10], sharp changes of ionic strength, temperature, and pH of the estuarine waters cause reactive metals such as iron (Fe), manganese (Mn), neodymium (Nd), or hafnium (Hf) to be removed from solution to a large extent (>50%) via salt-induced coagulation and precipitation of (nano-)particles and colloids (NPCs)[11,12], thereby reducing the fluxes of these metals to the Atlantic Ocean. However, inputs from diverse tributaries and suspended particle dissolution within the Amazon freshwater plume can partially compensate for the loss of rare earth elements and likely other elements in the

estuary[13]. The incomplete understanding of these processes limits the accuracy of trace element fluxes from the Amazon estuary to the Atlantic Ocean.

Radiogenic Nd and Hf isotopic compositions, expressed as $\varepsilon_{Nd}$ and $\varepsilon_{Hf}$, respectively (defined by equations in Methods) are sensitive tracers of the origin and mixing of water masses[14–16] enabled by fingerprinting of their continental source contributions with distinct isotopic signatures and their quasi-conservative behavior in seawater resulting in oceanic residence times of 300–1000 years and 250–7500 years, respectively[17–21]. While Nd isotopes weather largely congruently, Hf isotopes are strongly affected by incongruent weathering processes[20,22] and thus have widely been used to investigate the intensity and regimes of continental weathering and local to regional water mass mixing[23–25]. Therefore, $\varepsilon_{Nd}$ and $\varepsilon_{Hf}$ are important tracers of water mixing and seawater-particle interactions in estuaries and given that rivers drain different catchments, ambient seawater and particles are often characterized by distinct isotopic fingerprints. Rousseau et al.[13] used dissolved and particulate Nd concentrations ([Nd]) and $\varepsilon_{Nd}$ signatures to investigate seawater-particle interaction processes in the Amazon estuary and observed a slight increase of dissolved [Nd] accompanied

[1]GEOMAR Helmholtz Centre for Ocean Research Kiel, Kiel, Germany. [2]Department of Oceanography, Dalhousie University, Halifax, NS, Canada. [3]Woods Hole Oceanographic Institution, Woods Hole, MA, USA. ✉e-mail: axu@geomar.de

by a shift in $\varepsilon_{Nd}$ to less radiogenic values at mid to high salinities. These changes led them to conclude that the dissolution of (re)suspended particles releases Nd to the river plume and that particle dissolution in estuaries is an essential source term for the global marine Nd budget, possibly affecting other elements as well. However, particle dissolution includes both the Fe–Mn oxyhydroxide and silicate phases, which likely carry different isotopic signatures that were not distinguished previously. Furthermore, potential additional sources of Nd (and Hf) were not considered, such as the adjacent Pará River, whose dissolved [Nd] and $\varepsilon_{Nd}$ signatures were unknown but which discharges $6.6 \times 10^{11} m^3$ of freshwater annually into the Amazon estuary[26].

To investigate the contributions of all potential sources to the estuary and the Atlantic Ocean, we report the isotopic compositions of dissolved Nd and Hf together with those of particulate Nd, and the concentrations of dissolved rare earth elements and yttrium ([REY]) and Hf ([Hf]) in surface waters along the entire salinity (Sal) gradient of the Amazon estuary, including the Pará River outflow (Fig. 1). Samples were obtained in April–May 2018 during RV Meteor cruise M147, which was official process study Gapr11 of the international GEOTRACES program. The $\varepsilon_{Nd}$ and $\varepsilon_{Hf}$ signatures reveal significant dissolved Nd and Hf inputs from the Pará River to the outer Amazon estuary, with far-reaching implications for the global marine Nd and Hf budgets and potentially also for those of other trace elements.

## Results and discussion
### Variability of estuarine REY and Hf concentrations

Dissolved [Nd] and [Hf] of the Pará River endmember (Sal = 0.3) are 1036 pmol kg⁻¹ and 13.4 pmol kg⁻¹, respectively, and are thus significantly higher than 502 pmol kg⁻¹ and 12.3 pmol kg⁻¹ of the Amazon River freshwater endmember (Sal = 0.2) (Supplementary Data 1). These higher concentrations are likely a result of the overall lower pH of the Pará River pH of 6.2–7.4 (Amazon River: 6.8–7.3)[27] and/or feeding by tributaries from the mangrove forests, which is consistent with high trace metal export from the Amazonian mangrove forest areas[28–30]. In addition, parent rock characteristics and floodplain supply may play a role given that elevated REY concentrations in the waters exiting the floodplain have been observed in the Amazon Basin[31]. The dissolved [Nd] of Amazon River water sampled in April–May 2018 agrees well with

the values of 471–579 pmol kg⁻¹ reported for August 1989[32] but is substantially lower than the 850 pmol kg⁻¹ found in April 2008[13] documenting a dynamic mixing regime in the estuary and significant interannual variability, which may be related to biogeochemical changes in the floodplain[31].

The Amazon River and Pará River freshwaters exhibit characteristic middle REY (MREY) enrichment patterns after normalization to Post-Archean Australian Shale (PAAS)[31,33–35] (Fig. 2a–c). The REY patterns evolve towards light REY (LREY) depletion and heavy REY (HREY) enrichment signatures with increasing salinity and reach a typical seawater pattern with pronounced cerium (Ce) anomalies in the high-salinity zone (Sal > 25)[36] following large-scale removal of dissolved REY in the low-salinity zone (Fig. 2d, e).

Along the salinity gradient of all three transects, dissolved [Nd] and [Hf] decrease rapidly in the low-salinity zone (Sal < 6) due to the modification of the negative surface-charge of nanoparticles and colloids (NPCs) by seawater cations leading to flocculation and precipitation of river-borne NPCs[11,37,38]. The maximum removal of 90.8% and 95.0% for Nd and 87.3% and 82.5% for Hf is observed in the north Amazon and Pará transects, respectively (Fig. 2f, g). These observations demonstrate that Hf, similar to HREY[33], is removed less efficiently than Nd during estuarine mixing, likely due to the relatively higher free Nd ion content of river water. As a net effect of competing solution- and surface-complexation processes, the proportion of Nd in the colloidal fraction is higher[33] and hence more Nd than Hf is removed by coagulation, which is also consistent with the decrease in removal efficiency from LREY to HREY during estuarine mixing[32,39]. This is supported by recent data from the Congo River estuary, where at least 57% of the dissolved riverine Nd is removed during estuarine mixing (Sal < 23) but little or no estuarine removal of Hf has been observed based on the seasonal Congo River [Nd] and [Hf] data[34]. In the Hudson River estuary, an increase of [Hf] at salinities between 5 and 15 has been observed, possibly related to pore water diffusion, groundwater advection or release from resuspended particles[40]. To examine whether coagulation and removal of Nd and Hf continue with increasing salinity or Nd and Hf addition in the mid- to high-salinity zone occurs after initial removal, a second conservative mixing line (green dashed line in Fig. 3) was calculated between the water samples corresponding to the point of maximum removal

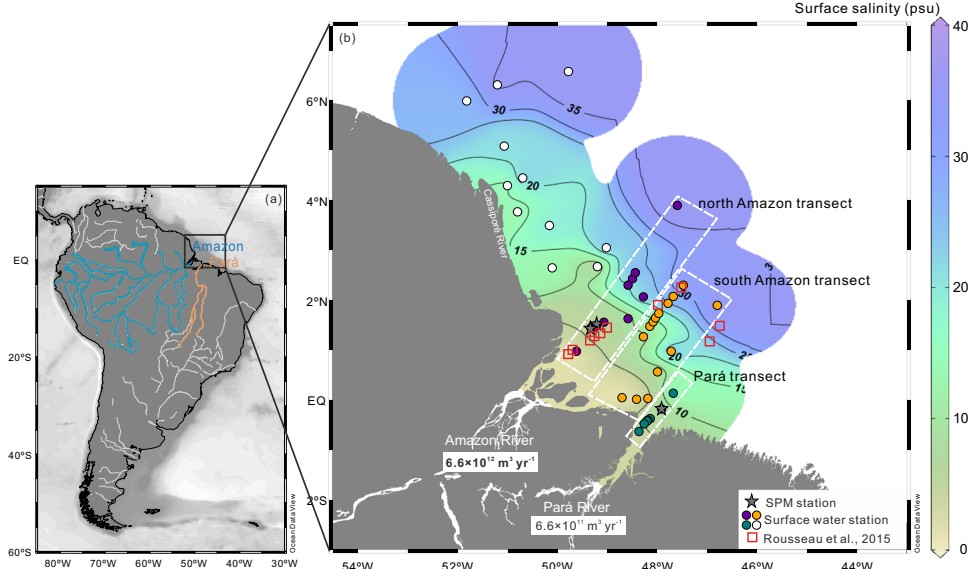

**Fig. 1 | Map of the study area showing sampling sites. a** Location of study area on the North Brazil continental shelf. **b** Sampling sites in the Amazon estuary. Stars represent stations with suspended particulate matter (SPM) sampling. The three transects across the Amazon and Pará River estuaries are highlighted by differently

colored circles and dashed rectangles. Open squares show the sampling stations of Rousseau et al.[13]. The mean annual freshwater discharges of the Amazon River and Pará River are indicated[10,26]. The map was created using Ocean Data View (https://odv.awi.de/)[79].

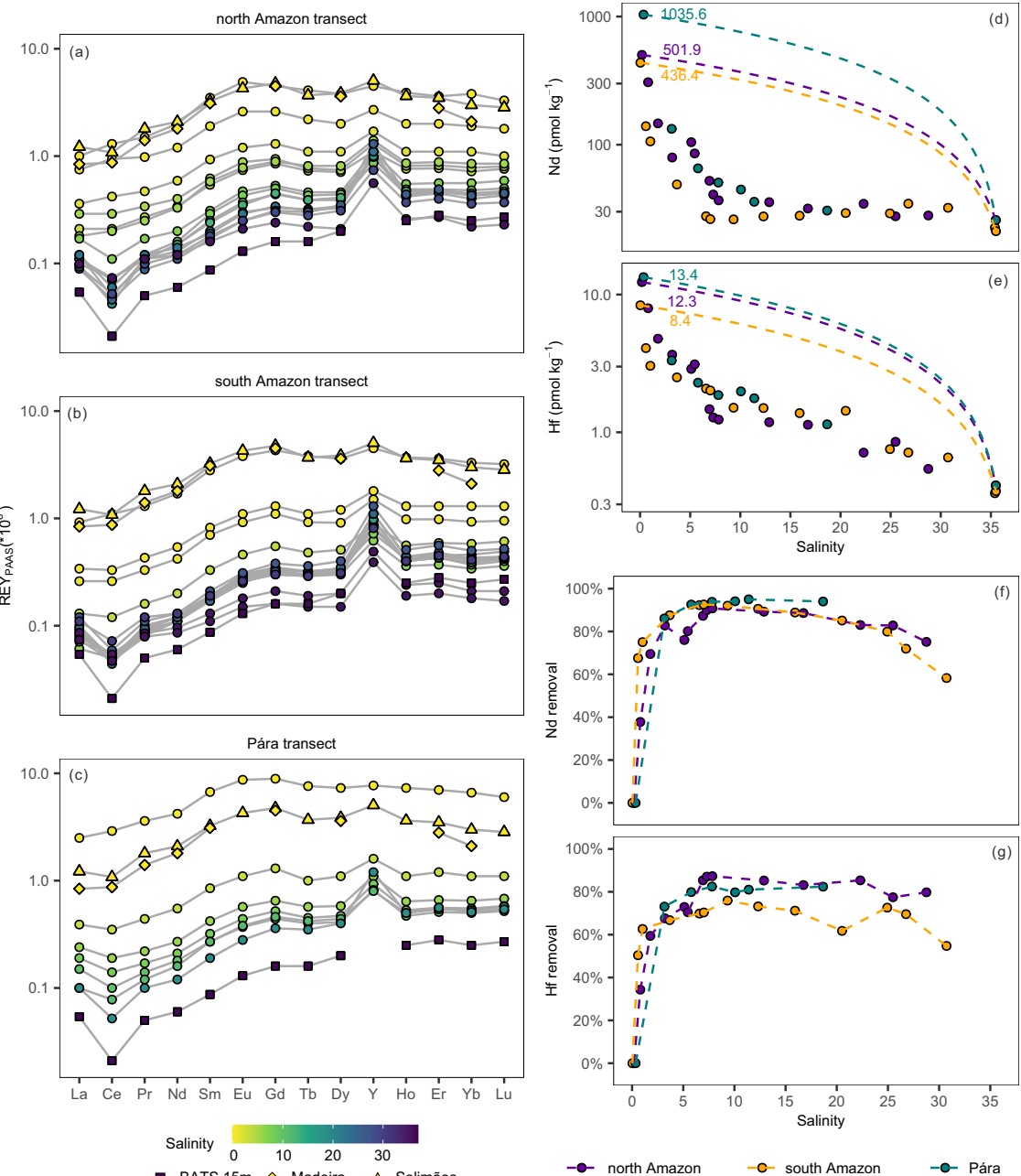

**Fig. 2 | Rare earth elements and yttrium (REY) patterns and Nd and Hf concentration ([Nd], [Hf]) against salinity in surface water of the three estuarine transects.** Post-Archean Australian Shale (PAAS) normalized REY patterns of surface water from the north Amazon transect (**a**), the south Amazon transect (**b**), and the Pará transect (**c**). Solimões river water[41] and Madeira river water[31] in the Amazon basin and BATS 15 m seawater (Sal: 36.5 psu) from the western North Atlantic[78] are shown for comparison. Distributions of dissolved [Nd] (**d**) and [Hf] (**e**) of surface waters of the three estuarine transects. The dashed lines represent calculated conservative mixing lines between the freshwater (Sal = 0.2–0.3) and seawater endmembers (Sal ≥ 35) in each transect. Variations in Nd and Hf removal percentage in surface water with respect to salinity (**f**, **g**), quantified using Eq. (3) in Methods. Salinity is given in psu.

percentage and the seawater endmembers. Interestingly, in the south Amazon transect, we observed elevated [Nd] and [Hf] compared to their second conservative mixing lines for salinities of 10–30 and 10–25, respectively (statistical significance $P < 0.01$ for Nd and $P = 0.06$ for Hf, $t$-test) (Fig. 3c, d) with a significant increase in [Nd] from 26 pmol kg⁻¹ to 35 pmol kg⁻¹, documenting the contribution of a third endmember other than Amazon River and Atlantic seawater.

## $\varepsilon_{Nd}$ and $\varepsilon_{Hf}$ distributions across the Amazon plume

The dissolved $\varepsilon_{Nd}$ and $\varepsilon_{Hf}$ signatures of Amazon River freshwater are −9.4 ± 0.2 and +1.8 ± 0.9, respectively, while those of Pará River freshwater are markedly less radiogenic, reaching −14.1 ± 0.2 and -4.1 ± 0.6,

respectively. These Pará River signatures are likely associated with higher contributions from weathering of the cratonic Shield, whose parent rock and suspended particulate matter (SPM) $\varepsilon_{Nd}$ signals predominantly range from −16 to −24[41–43]. The $\varepsilon_{Nd}$ of the Amazon River endmember in our study agrees well with previously measured values of 9.2 ± 0.4[44], −8.9 ± 0.5[45] and −8.8 ± 0.2[13]. The $\varepsilon_{Nd}$ values of the residual Amazon River SPM (after removal of Fe–Mn oxyhydroxides through leaching) are consistent at −11.8 ± 0.03 (±1 standard deviation, SD, $n = 5$, open red triangles in Fig. 4a) and less radiogenic than -10.7[13] or -10.3[41] obtained for bulk SPM samples from a similar section (Fig. 1). In contrast, the $\varepsilon_{Nd}$ signatures of the Fe–Mn oxyhydroxide phase of SPM that would most likely dissolve and release Nd to the plume along the north Amazon

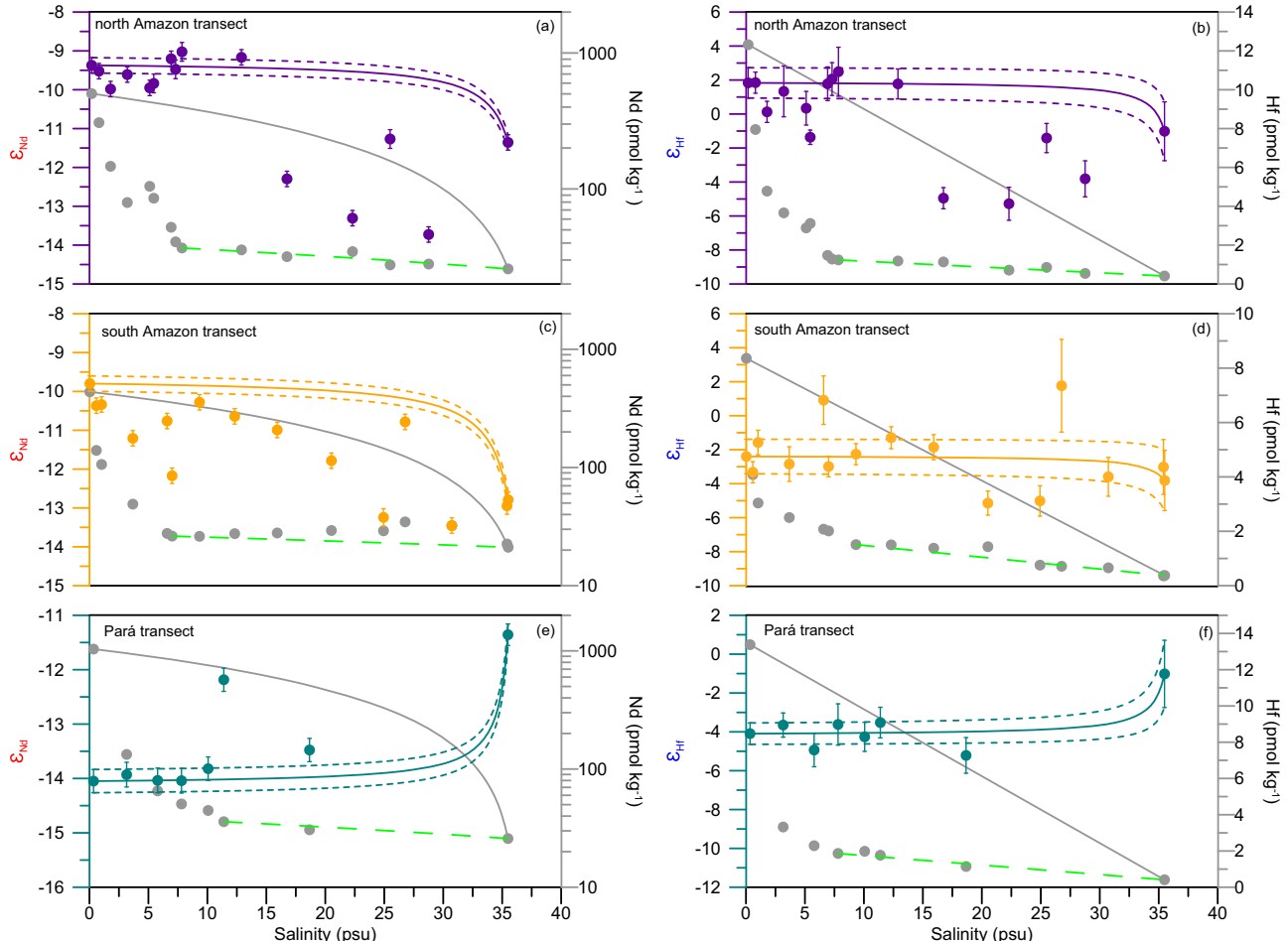

**Fig. 3 | Nd and Hf isotopic composition ($\varepsilon_{Nd}$, $\varepsilon_{Hf}$) and concentration distributions along the salinity gradients of the three estuarine transects.** Distributions of dissolved concentrations and isotopic compositions of Nd and Hf along the north Amazon transect (**a**, **b**), the south Amazon transect (**c**, **d**) and the Pará transect (**e**, **f**). The solid lines in each panel represent the predicted conservative mixing lines between the river water and seawater endmembers for concentrations and isotopic compositions, respectively. The dashed lines above and below the corresponding solid lines reflect the uncertainties of the estimations of the $\varepsilon_{Nd}$ and $\varepsilon_{Hf}$ endmember values. Error bars correspond to the 2 standard deviations of the $\varepsilon_{Nd}$ and $\varepsilon_{Hf}$ measurements. The second conservative mixing line (green dashed line) for Nd and Hf is defined by water samples corresponding to the maximum removal percentage and the seawater endmembers.

transect (−8.4 ± 0.2 ‒ −8.1 ± 0.2) (open red squares in Fig. 4a) are slightly more radiogenic and similar to the dissolved $\varepsilon_{Nd}$ signature of Amazon River water. These values are also intermediate between the $\varepsilon_{Nd}$ signatures of the Solimões (−7.1) and Madeira (−10.0), suggesting that the Fe–Mn oxyhydroxide fraction of SPMs sampled in the north Amazon transect mainly reflects the isotopic signals of the Amazon River that are governed by Andean tributaries[41]. The $\varepsilon_{Nd}$ signature of Fe–Mn oxyhydroxides in the Pará transect is −10.6 ± 0.2 (open green squares in Fig. 4a) and thus less radiogenic than those of the north Amazon transect.

Figure 3 shows the evolution of $\varepsilon_{Nd}$ and $\varepsilon_{Hf}$ in the dissolved phase along the Amazon estuary surface salinity gradient. Throughout the low-salinity zone (Sal: 0 ‒ 10) of the north Amazon and Pará transects, $\varepsilon_{Nd}$ and $\varepsilon_{Hf}$ values are close to the conservative mixing lines, although large amounts of dissolved Nd and Hf are clearly removed from solution, supporting the efficient removal of dissolved REY and Hf without alteration or fractionation of the radiogenic isotope signatures in the estuarine waters (Fig. 3a, b, e, f). In the mid- to high-salinity region of the south Amazon transect, where elevated dissolved [Nd] and [Hf] occur, we observe a gradual decrease in $\varepsilon_{Nd}$ and $\varepsilon_{Hf}$ signatures to −13.7 ± 0.2 and −5.3 ± 1.0, respectively. Conservative mixing between the Amazon River and open Atlantic waters alone is insufficient to explain these gradients and values in the Amazon estuary, let alone the increase of dissolved [Nd] in the mid-salinity region. In previous

studies, this shift was attributed to release from sediments or suspended particles[13,32], but our data support an alternative explanation.

## Source of elevated dissolved [Nd] and [Hf] in the Amazon estuary

The most important observation of our study is the increase in dissolved [Nd] and [Hf] at mid to high salinities accompanied by a shift to highly unradiogenic $\varepsilon_{Nd}$ and $\varepsilon_{Hf}$ values. This pattern could either be due to release from SPM, as suggested previously[13], or the result of additions from other dissolved Nd and Hf sources. The labile phases of SPM are mainly Fe–Mn oxyhydroxides that tend to dissolve and release Nd to the water and are sensitive to pH and redox conditions[46]. Dissolution of 0.5%–8.0% and 0.3%–5.1% of the labile Fe–Mn oxyhydroxide phases in particles from the Amazon and Pará, respectively, or partial dissolution of residual SPM (0.1%–1.1% and 0.1%–0.9%, respectively) could raise the dissolved [Nd] from the observed 26 pmol kg⁻¹ to 35 pmol kg⁻¹. However, based on our measured radiogenic Nd isotope values, the Amazon River freshwater-seawater endmember mixing combined with particle dissolution cannot account for the unradiogenic $\varepsilon_{Nd}$ signatures (Fig. 4a) observed in the south Amazon transect (Supplementary Fig. 1). This is also supported by consistent REY patterns (Fig. 2b) in the south Amazon transect across the mid to high-salinity gradient of the surface estuarine waters suggesting water mass mixing without REY addition from particle release, which would result

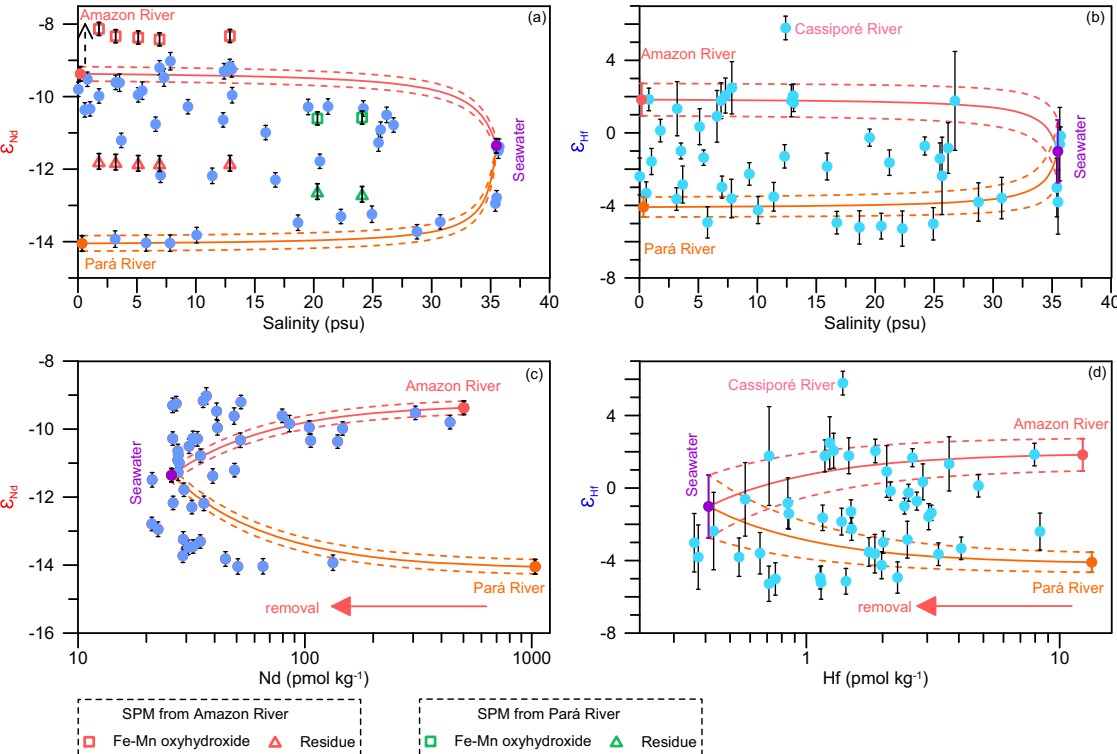

**Fig. 4 | Three-endmember mixing plots of Nd and Hf isotopes ($\varepsilon_{Nd}$, $\varepsilon_{Hf}$) in the Amazon estuary.** Three-endmember mixing relationships of $\varepsilon_{Nd}$ (**a**) and $\varepsilon_{Hf}$ (**b**) calculated based on the $^{143}Nd/^{144}Nd$ and $^{176}Hf/^{177}Hf$ ratios with corresponding Nd concentrations ([Nd]) and Hf concentrations ([Hf]) and their salinities of three dissolved sources. The open squares and triangles represent the $\varepsilon_{Nd}$ signatures of the Fe–Mn oxyhydroxide fraction and of the total residual suspended particulate matter (SPM) from the Amazon River and Pará River, respectively. $\varepsilon_{Nd}$ (**c**) and $\varepsilon_{Hf}$ (**d**) distributions in the Amazon estuary against [Nd] and [Hf], respectively, showing the large-scale estuarine removal of Nd and Hf. The solid lines represent conservative mixing between the freshwater and seawater endmembers. The dashed lines above and below the corresponding solid line represent uncertainties in the estimation of the $\varepsilon_{Nd}$ and $\varepsilon_{Hf}$ endmember values. Error bars correspond to the 2 standard deviations of the $\varepsilon_{Nd}$ and $\varepsilon_{Hf}$ measurements.

in flatter REY patterns with indiscernible LREY depletion (Supplementary Fig. 2). To further constrain potential sedimentary Nd and Hf sources, seven near-bottom water samples recovered in the continental shelf area of the Amazon estuary (Supplementary Fig. 3) were measured. Their mean $\varepsilon_{Nd}$ and $\varepsilon_{Hf}$ signatures of are $-11.0 \pm 1.2$ ($\pm 1$ SD, $n = 7$) and $1.0 \pm 1.7$ ($\pm 1$ SD, $n = 7$), respectively, excluding bottom supply as a significant source of the observed unradiogenic surface water $\varepsilon_{Nd}$ and $\varepsilon_{Hf}$ signatures. Particle dissolution/particle-seawater interaction (i.e., boundary exchange processes) may still occur but will be restricted to the bottom layer below the freshwater plume on the continental shelf and deep-sea fan. Based on the above evidence, admixture of Pará River water, which has the highest dissolved [Nd] and [Hf] (1036 pmol kg$^{-1}$ and 13.4 pmol kg$^{-1}$, respectively) and least radiogenic $\varepsilon_{Nd}$ and $\varepsilon_{Hf}$ signatures ($-14.1 \pm 0.2$, $-4.1 \pm 0.6$, respectively, Fig. 4) is the most likely explanation for the shift in isotopic signatures to highly unradiogenic values along the salinity gradient of the Amazon surface water plume. This is supported by a box model (Supplementary Figs. 4 and 5), showing that admixture of Pará River water can indeed shift the $\varepsilon_{Nd}$ and $\varepsilon_{Hf}$ in the outer Amazon estuary to values of $-13.9 \sim -13.7$ and $-4.1 \sim -3.6$, respectively, which are identical within error to the measured values, indicating an additional sedimentary source is not required to explain the data. This possibility had not been considered previously because data for the Pará River were not available. Examining the spatial $\varepsilon_{Nd}$ and $\varepsilon_{Hf}$ distributions across the estuarine surface waters clearly reveals this impact of the Pará River on the Nd and Hf signals of the Amazon plume (Fig. 5a, b).

A three-endmember mixing model is applied to quantify the fraction of Nd and Hf added to the Amazon plume from the Pará River along the estuarine salinity gradient since the efficient removal process does not alter the $\varepsilon_{Nd}$ and $\varepsilon_{Nd}$ signatures. The properties of the three

major dissolved Nd endmembers (Amazon River freshwater, Pará River freshwater and Atlantic seawater) are compiled in Table 1. To compare the relative contribution of isotope signatures from the Amazon and Pará rivers, the riverine Nd and Hf proportion originating from the Pará River (named Pará riverine Nd fraction and Hf fraction) in each water sample was calculated and displayed numerically (Fig. 5c, d). In the inner Amazon River estuary (Sal <20), the Amazon River dominates the isotope signatures of estuarine waters by supplying 55%–100% and 54%–100% riverine Nd and Hf, respectively, as indicated by more radiogenic $\varepsilon_{Nd}$ and $\varepsilon_{Hf}$ signatures. In contrast, the Pará River component exceeds that of the Amazon River and dominates the isotope signature of estuarine waters in the southern and outer estuary (20 < Sal < 35) accounting for 45%–100% and 46%–100% of the riverine Nd and Hf input, respectively. The large and previously overlooked dissolved Nd and Hf contributions from the Pará River are supported by the regional coastal circulation and satellite images of mud distribution in the estuary (Supplementary Fig. 6), as well as high concentrations of other trace metals (Fe, nickel, cobalt, titanium, aluminum, zinc, lead) in the Pará River[27–29]. Therefore, the Pará River is an essential source of micronutrients to the Amazon estuary and to the western Atlantic and thus needs to be considered in future studies on the budget of trace elements of the western Atlantic Ocean.

## Implications for global riverine dissolved Nd and Hf fluxes

Accurate oceanic budgets of Nd and Hf are needed to reliably apply Nd and Hf isotopes as tracers of water mass sources and their mixing in the modern ocean and as proxies for past changes in global ocean circulation. In the modern ocean, Hf sources have not yet been precisely constrained but include dust/particle release[47,48], riverine inputs[20,34,49,50] and hydrothermal systems[51,52]. The riverine flux of Hf to

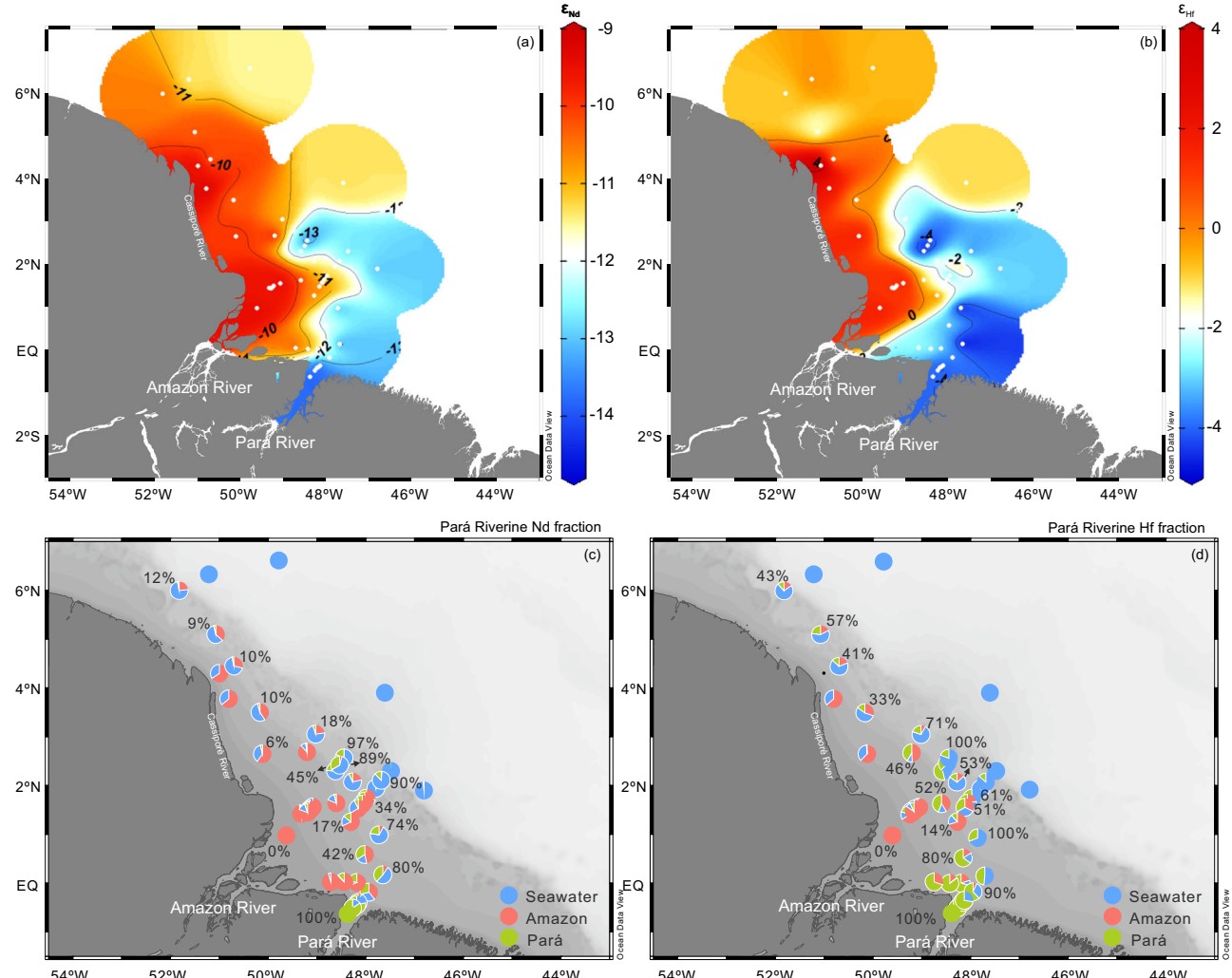

**Fig. 5 | Distributions of Nd and Hf isotopes ($\varepsilon_{Nd}$, $\varepsilon_{Hf}$) and water fractions of the Pará River, the Amazon River and Atlantic seawater in the Amazon estuary.** Distributions of $\varepsilon_{Nd}$ (**a**) and $\varepsilon_{Hf}$ (**b**) in the estuary. Pará riverine Nd fraction (**c**) and Hf fraction (**d**) defined by Eq. (8) in Methods and displayed numerically. Fractions of Pará River water, Amazon River water and Atlantic seawater in the Amazon estuary are represented as pie charts calculated by Eqs. (4), (5), (6), and (7). Figures were produced using Ocean Data View (https://odv.awi.de/)[79].

**Table 1 | Dissolved Nd and Hf concentrations ([Nd], [Hf]) and isotopic signatures ($\varepsilon_{Nd}$, $\varepsilon_{Hf}$) of the different endmembers**

| | Transect ID | Sal psu | [Nd] pmol kg⁻¹ | $\varepsilon_{Nd}$ | ¹⁴³Nd/¹⁴⁴Nd | [Hf] pmol kg⁻¹ | $\varepsilon_{Hf}$ | ¹⁷⁶Hf/¹⁷⁷Hf |
|---|---|---|---|---|---|---|---|---|
| Amazon River | North Amazon | 0.2 | 501.9 | −9.4 | 0.512158 | 12.3 | 1.8 | 0.282837 |
| Pará River | Pará | 0.4 | 1035.6 | −14.1 | 0.511918 | 13.4 | −4.1 | 0.282669 |
| Atlantic seawater | — | 35.5 | 25.8 | −11.4 | 0.512056 | 0.4 | −1.0 | 0.282756 |
| Labile Fe–Mn oxyhydroxide | North Amazon | — | — | −8.3 ± 0.1 | 0.512211 | — | — | — |
| | Pará | — | — | −10.6 ± 0.02 | 0.512094 | — | — | — |
| Residual SPM | North Amazon | — | — | −11.8 ± 0.03 | 0.512031 | — | — | — |
| | Pará | — | — | −12.7 ± 0.05 | 0.511987 | — | — | — |

The $\varepsilon_{Nd}$ of Fe–Mn oxyhydroxide and residual SPM are mean values in each transect. The standard deviation for these values is calculated as the mean value over 5 and 2 samples in the north Amazon transect and the Pará transect, respectively.

the oceans is not well quantified with remarkably low [Hf] in seawater resulting in limited data availability and due to variable Hf removal during estuarine mixing[34,40]. Therefore, it remains difficult to assess the importance of riverine Hf fluxes for the global Hf budget. There is considerably more information available on Nd inputs to the ocean via (1) dust with an estimated flux of 2 - 4 × 10⁸ g yr⁻¹ Nd[19,21,53], and (2) rivers

with an estimated flux of 3−5 × 10⁸ g yr⁻¹ Nd[19,21]. However, the above Nd source fluxes are not sufficient to balance both [Nd] and $\varepsilon_{Nd}$ distributions in the global ocean, resulting in a substantial deficit of Nd inputs of 5−11 × 10⁹ g yr⁻¹ estimated by models[19,21,53,54]. Possible additional Nd sources include submarine groundwater discharge[55], benthic sediment fluxes[56] and release via particle dissolution[13]. However, the average

riverine [Nd] of 284 pmol kg⁻¹ and 70% Nd estuarine removal, used in most models[19,21,53–55] were calculated based on the relationship between [Nd] and the concentrations of Ca and Na across the salinity gradient in two estuaries, respectively[57,58]. In view of higher than previously assumed [Nd] observed in our study and the low Nd removal (~57%) recently reported for the Congo River estuary[34] (the second largest river globally in terms of volume discharge), the flux of riverine Nd needs to be re-examined to improve our understanding of the sources of Nd in the modern ocean.

We have compiled dissolved [Nd] measurements of 49 globally distributed rivers together with their discharge and available pH values (48/49) (Supplementary Data 2) and recalculated the discharge-weighted mean dissolved riverine [Nd] ($997 \pm 36$ pmol kg⁻¹, $\pm 1$ standard error, SE) (see Methods) as a basis for further evaluation in a global context. A correlation between the pH of rivers and the dissolved riverine [Nd] has been noticed since the earliest studies and was attributed to solution chemistry[6,33]. The strong inverse correlation between pH and log [Nd] ($R^2 = 0.65$, $P < 0.01$, $n = 48$, Fig. 6a) is remarkable given the likely secondary control of rock type and the amount of colloidal material in different rivers[33] and allows us to predict riverine dissolved [Nd] based on the large pH dataset available for global river waters ($n = 582$) (GEMStat[59,60] and GLORICH[61]) (Supplementary Data 2). The resulting predicted weighted mean dissolved riverine [Nd] is $943 \pm 64$ pmol kg⁻¹ ($\pm 1$SE, $n = 582$) (Fig. 6b), which is similar to the observed discharge-weighted mean river dissolved [Nd] estimated above, but more than 3–6 times the concentrations of 284 pmol kg⁻¹ and 146 pmol kg⁻¹ previously used in models[19,21,53–55]. Considering a global river discharge ($133.1 \times 10^4$ m³ s⁻¹)[57], a global riverine flux of $5.7 \times 10^9$ g Nd yr⁻¹ is calculated, which is (nearly) identical to previous estimates in two studies that took the [Nd] of only 40 or 21 rivers into account ($5.4$ or $5.7 \times 10^9$ g Nd yr⁻¹)[62,63]. Using a larger global dataset ($n = 582$), our study confirms the few previous estimates of the dissolved riverine [Nd] flux while increasing data coverage by over 10-fold. The net dissolved Nd flux from rivers to the oceans is, however, controlled by the efficiency of the estuarine removal processes. A discharge-weighted mean maximum Nd removal percentage of $85 \pm 4\%$ ($\pm 1$SE, $n = 12$) is calculated from published data (12 estuarine transects) (Supplementary Data 2) (see Methods). As revealed by mixing experiments[38] and a lower Nd removal percentage (~50%) in the Mississippi River estuary attributed to strong aqueous complexation of REY with natural organic ligands and carbonate ions[64], we find that maximum Nd removal is closely related to dissolved organic carbon concentration ([DOC]) based on the 12 available estuarine transects. A linear correlation ($R^2 = 0.52$, $P < 0.01$) between existing observations (Fig. 6c) predicts a discharge-weighted mean maximum Nd removal percentage of $74 \pm 1\%$ ($\pm 1$SE, $n = 211$) calculated based on the discharge-weighted mean [DOC] of over 211 rivers from a larger global dataset ($n = 211$) (GEMStat[59,60] and GLORICH[61]) (Fig. 6d). This result is consistent with the previously used 70% removal employed in models[19,21,53–55]. Therefore, the revised net dissolved riverine Nd flux to the global oceans is $1.5 \times 10^9$ g Nd yr⁻¹ (Table 2). This does not include any flux from river derived sediments as observed on continental shelves[24,34], but is still 3 to 5 times higher than the estimated global dust Nd input, and accounts for 17 to 27% of the global Nd input to the oceans estimated in models[19,21,53].

The revised global riverine dissolved Nd flux implies that this flux was previously significantly underestimated. The revised riverine Nd flux of $5.7 \times 10^9$ g yr⁻¹ and maximum removal percentage of 74% should be implemented for models of Nd cycling in the future (Table 2). There are still many limitations related to the small available dataset of [Nd], pH values and [DOC]. Based on the existing data, riverine [DOC] seasonally variable but the scarcity of such data complicates constraining the relationship between [DOC] and the maximum Nd removal percentage. In addition, Nd removal is influenced by the content of NPCs (e.g., colloidal Fe and Mn oxides)[38], which cannot be estimated globally here due to the limited available data. Therefore, more studies on rivers during both wet and dry seasons are needed to better constrain annual riverine Nd fluxes and to improve our understanding of the global Nd budget. Our findings also suggest that future work should focus on the role of rivers with low pH and high [DOC], which can contain high trace element concentrations (e.g., Fe, Mn)[65] and display low estuarine removal[38,64], as important sources of global trace metal fluxes to the surface oceans.

## Methods

### Sample collection and treatment

48 water samples along the entire salinity gradient (0 to >35) in the Amazon River estuary, the Pará River estuary and nearby regions of the Brazilian continental shelf (Fig. 1) were collected during RV Meteor cruise M147 (29 April to 20 May 2018; official process study GApr11 of the international GEOTRACES program). Surface water samples were collected either with a towed-fish or a Conductivity-Temperature-Depth (CTD) rosette equipped with 24-Niskin bottles. With the towed-fish the water samples were recovered at 2 to 3 m water depth and the bottles of the CTD rosette were closed at the same depth immediately below the surface to sample the uppermost freshwater layer. For each sample, 20 to 40 L were transferred into acid-cleaned 20 L plastic cubic containers and filtered through 0.45 µm Nucleopore filters within a few hours after collection. All samples were acidified on board to pH ~2 using concentrated ultrapure distilled HCl and stored at room temperature for further treatment in the clean room laboratory at GEOMAR.

SPM samples were collected from the large amount of settled particulate material in the 20 L CTD sample plastic cubitainers of the low-salinity samples. Thus the finest fraction retained on the 0.45 µm Nucleopore filters has not been analyzed, which accounted for $3.1 \pm 1.0\%$ (1 SD, $n = 5$) of the bulk SPM sample mass calculated based on the filter weights and SPM content of SPM samples. The SPM was rinsed into smaller acid-cleaned containers using MQ water and excess water removed by siphoning after the SPM had visually settled. In the laboratory, the SPM was freeze dried at −52 °C and then homogenized before further treatment.

### Fe−Mn oxyhydroxide extraction and alkaline fusion

Approximately 0.2 to 1 g of SPM was treated with a diluted reductive solution consisting of 0.005 M hydroxylamine hydrochloride/1.5% acetic acid/0.03 M Na-EDTA (sodium-ethylenediaminetetraacetate) solution buffered to pH 4 with NaOH (sodium hydroxide) for 10 minutes, following Huang et al.[66] for extracting the labile Fe−Mn oxyhydroxide fraction. Total procedural blanks were below 15 pg for Nd ($n = 2$) and are hence negligible. The reproducibility was monitored by leaching marine sediment reference material MESS-2 ($n = 3$). The SPMs were then treated with strong reductive solution consisting of 0.05 M hydroxylamine hydrochloride/15% acetic acid/0.03 M Na-EDTA solution buffered to pH 4 with NaOH overnight to completely remove residual Fe−Mn oxyhydroxides, following the method applied in Gutjahr et al.[67] In this study, the labile Fe−Mn oxyhydroxide phase data we measured refers specifically to the fraction extracted by the diluted reductive solution. The residual SPM samples were dried at low temperature (<45°C) in an oven and homogenized prior to alkaline fusion following Bayon et al.[68] The accuracy and reproducibility of the fusion technique was monitored by processing reference materials with each batch of SPM samples including marine sediment MESS-2 ($n = 3$), USGS reference material BHVO-2 ($n = 3$) and AGV-2 ($n = 3$). Total procedural blanks were below 80 pg for Nd ($n = 4$) are therefore negligible.

### Neodymium and hafnium isotope analyses

Filtered water samples for $\varepsilon_{Nd}$ and $\varepsilon_{Hf}$ analysis were pre-concentrated by iron (Fe) co-precipitation. To remove most of this Fe, the samples were treated with pre-cleaned di-ethyl ether[16]. The REY and Hf of water

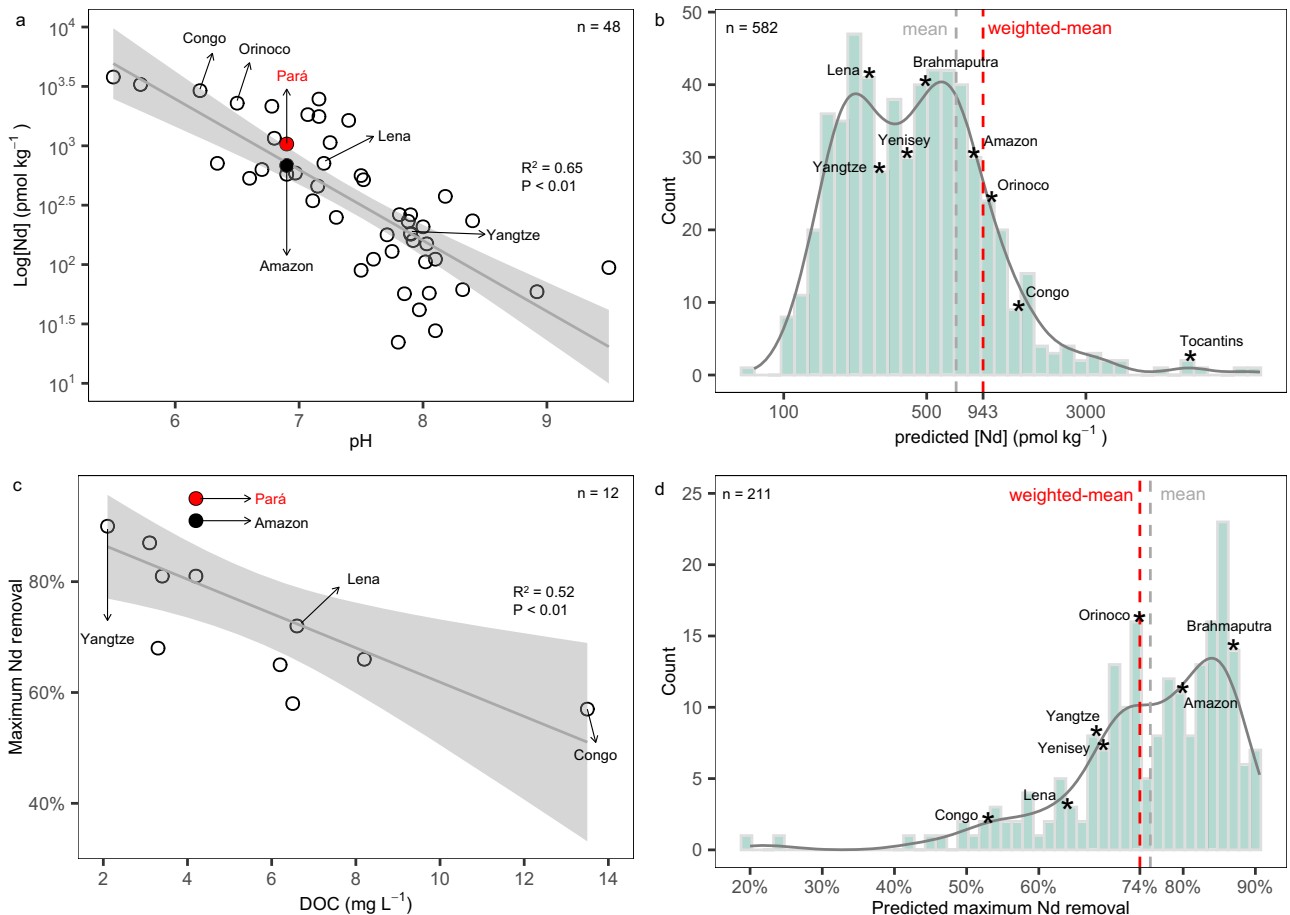

**Fig. 6 | Relationships between river pH and Nd concentration ([Nd]), and between dissolved organic carbon concentration ([DOC]) and maximum Nd removal percentage in estuaries. a** Relationship between river pH and dissolved log [Nd] calculated from 48 rivers. **b** Predicted discharge-weighted mean [Nd] based on the available global dataset (*n* = 582). **c** Relationship between [DOC] and maximum Nd removal percentage calculated from 12 rivers. **d** Predicted discharge-weighted mean maximum Nd removal percentage based on the available global dataset (*n* = 211). Some large rivers have been flagged in panels **b** and **d** to show the impact of these highly weighted rivers on the calculations of global discharge-weighted mean [Nd] and estuarine removal percentage.

and SPM samples were separated from matrix elements using cation exchange chromatography (AG® 50W-X8, 1.4 mL, 200–400 µm) following the scheme of Stichel et al.[16]. Nd was further separated from the other REY for isotope measurements using Eichrom® LN-Spec resin (2 mL, 50–100 µm) following the procedure of Pin and Zalduegui[69].

## Table 2 | Comparison of optimized Nd parameters between literatures and our study

| | Previously used | Revised |
|---|---|---|
| | | Weighted mean, 1SD, 1SE, 95% CI |
| [Nd]$_{observed}$ (pmol kg$^{-1}$) | 146, 284 | 997, 860, 36, [927, 1067] |
| [Nd]$_{predicted}$ (pmol kg$^{-1}$) | — | 943, 1534, 64, [818, 1067] |
| [Removal] $_{observed}$ (%) | 70 | 85, 13, 4, [78, 93] |
| [Removal]$_{predicted}$ (%) | — | 74, 9, 1, [73, 76] |
| Gross river Nd flux (g yr$^{-1}$) | 0.9–1.7 × 10$^9$ | 5.7 × 10$^9$ |
| Net river Nd flux (g yr$^{-1}$) | 0.3–0.5 × 10$^9$ | 1.5 × 10$^9$ |
| Global Nd budget (g yr$^{-1}$) | 6–9 × 10$^9$ | |

Previously used [Nd]$_{observed}$, [Removal]$_{observed}$, gross, net river Nd flux and global Nd budget are based on previous studies[19,21,53–55]. The standard deviation (1SD), standard error (1SE) and 95% confidence interval (CI) of the calculations are listed below, and a detailed uncertainty assessment is included in the Supplementary Information.

The Hf was further purified by eluting titanium (Ti) and zirconium (Zr) using Bio-Rad AG® 1-X8 resin (2 mL, 200–400 µm) following Münker et al.[70]. The $^{143}Nd/^{144}Nd$ ratios were measured on a Neptune Plus MC-ICP-MS at GEOMAR and were corrected for instrumental mass bias to $^{146}Nd/^{144}Nd = 0.7219$ and to $^{142}Nd/^{144}Nd = 1.141876$ following the approach of Vance and Thirlwall[71]. The $^{143}Nd/^{144}Nd$ ratios of all samples were normalized to bracketing analyses of JNdi-1 standard with a value of 0.512115[72]. The $^{176}Hf/^{177}Hf$ ratios of the samples measured on the Neptune Plus MC-ICP-MS were corrected for instrumental mass bias to $^{179}Hf/^{177}Hf = 0.7325$ applying an exponential mass fractionation law and standard JMC 475 values were within uncertainty of the accepted value of 0.28216[73]. The total procedural laboratory blanks for water samples (n = 6) were negligible at <29 pg for Nd and 5–24 pg for Hf compared to sample sizes of 10–20 ng (Nd) and 2–10 ng (Hf) for isotope measurements. Secondary standard solution NIST 3135a and USGS reference material NOD-A-1 were run with water samples and USGS reference material AGV-2 and BHVO-2 were run with SPM samples to check the accuracy and external reproducibility of the procedure for Nd isotope measurements (Supplementary Data 1). For Hf isotope measurement, an internal laboratory standard solution (CGHF1, Inorganic Ventures®) and USGS reference material NOD-A-1 were measure for monitoring external reproducibility. The external reproducibility of the Nd and Hf isotope measurements of water samples was determined using standard solutions with concentrations matching those of the measured samples in the range of 0.20−0.24 $\varepsilon_{Nd}$ units (2 SD) and 0.45−2.73 $\varepsilon_{Hf}$

units (2 SD), respectively, while it was 0.18–0.21 $\varepsilon_{Nd}$ units (2 SD) for measurements of particle samples. The 2 SD of the secondary standard NIST 3135a and CGHF1 are employed to illustrate the reproducibility of measured $\varepsilon_{Nd}$ and $\varepsilon_{Hf}$ in all figures and are shown in Supplementary Data 1. $\varepsilon_{Nd}$ and $\varepsilon_{Hf}$ are defined by the Eqs. (1) and (2), respectively:

$$\varepsilon_{Nd} = \left( \frac{(^{143}Nd/^{144}Nd)_{sample}}{(^{143}Nd/^{144}Nd)_{CHUR}} - 1 \right) \times 10^4 \tag{1}$$

$$\varepsilon_{Hf} = \left( \frac{(^{176}Hf/^{177}Hf)_{sample}}{(^{176}Hf/^{177}Hf)_{CHUR}} - 1 \right) \times 10^4 \tag{2}$$

where the $^{143}Nd/^{144}Nd$ and $^{176}Hf/^{177}Hf$ ratios of CHUR (Chondritic Uniform Reservoir) are 0.512638[74] and 0.282785[75], respectively.

### Neodymium and hafnium concentration analyses

For Nd and Hf concentration measurements of water samples, 50 mL (low-salinity samples) to 1 L (high-salinity samples) water sample aliquots were spiked with pre-weighed $^{150}Nd$ and $^{180}Hf$ spikes, pre-concentrated using Fe co-precipitation, and purified on a AG® 50W-X8 column following the scheme of Rahlf et al.[34]. Hf cuts were dissolved in 0.1 M HF for further purification using AG® 1-X8 resin (1.6 mL, 200–400 µm) to elute Ti, Zr and tungsten (W) following a procedure of Sahoo et al.[76]. The isotope dilution measurements of the Nd and Hf concentrations based on $^{150}Nd/^{144}Nd$ and the $^{178}Hf/^{180}Hf$ ratios were carried out on a Nu Plasma MC-ICP-MS. External reproducibility (2 SD)

$$\frac{f_{Ama} \times [Nd]_{Ama} \times (^{143}Nd/^{144}Nd)_{Ama} + f_{Pará} \times [Nd]_{Pará} \times (^{143}Nd/^{144}Nd)_{Pará} + f_{Atl} \times [Nd]_{Atl} \times (^{143}Nd/^{144}Nd)_{Atl}}{f_{Ama} \times [Nd]_{Ama} + f_{Pará} \times [Nd]_{Pará} + f_{Atl} \times [Nd]_{Atl}} = (^{143}Nd/^{144}Nd)_{sample} \tag{6}$$

$$\frac{f_{Ama} \times [Hf]_{Ama} \times (^{176}Hf/^{177}Hf)_{Ama} + f_{Pará} \times [Hf]_{Pará} \times (^{176}Hf/^{177}Hf)_{Pará} + f_{Atl} \times [Hf]_{Atl} \times (^{176}Hf/^{177}Hf)_{Atl}}{f_{Ama} \times [Hf]_{Ama} + f_{Pará} \times [Hf]_{Pará} + f_{Atl} \times [Hf]_{Atl}} = (^{176}Hf/^{177}Hf)_{sample} \tag{7}$$

was better than 0.6% for Nd and better than 1.2% for Hf according to repeated treatment and measurement of the same sample ($n = 5$).

### Rare earth elements and yttrium concentration analyses

All REY were pre-concentrated offline using a SeaFAST system (model M5 from Elemental Scientific) following a method updated from Hathorne et al.[77]. Using the new system, 12 mL of acidified water sample was loaded precisely on the column using a fifth syringe pump and after the matrix was washed away, the REY were eluted with 400 uL of 1.5 M $HNO_3$. Before pre-concentration, every blank, reference material and water sample (pH-2) was spiked with 12 µL of thulium solution (10 ng g⁻¹) to monitor yields, which were typically 99.9 ± 5.9% (± 1 SD, $n = 58$). Before analysis on a Thermo Element XR ICP-MS coupled with a CETAC "Aridus 2" desolvating nebulizer, all samples were diluted with 200 uL of 0.1% $HNO_3$ containing 10 ng g⁻¹ Re as an internal standard during measurement and to account for any sample evaporation since the pre-concentration. The use of the desolvating nebulizer increases sensitivity and also decreases oxide formation, which was monitored with element solutions of barium (Ba), Ce, praseodymium (Pr) + Nd, and samarium (Sm) + europium (Eu) + gadolinium (Gd) + terbium (Tb) at the start of each analytical session. Oxide formation was generally <0.01 ± 0.003% ($n = 3$, ±1 SD) for Ba, <0.05 ± 0.01% ($n = 3$, ±1 SD) for Ce, <0.04 ± 0.01% ($n = 3$, ± 1 SD) for Pr+Nd and <0.04 ± 0.13% ($n = 3$, ±1 SD) for the MREY. Certified natural river and estuarine water reference materials (NRCC SLRS-6 and SLEW-3) and GEOTRACES inter-calibration samples BATS 15 m and 2000 m[78] were pre-concentrated like the samples and measured to monitor external

reproducibility and accuracy. Mean values and 2 SD for the reference material measurements are given in Supplementary Data 1.

### Nd and Hf removal and three-endmember model calculations

Nd and Hf removal percentage are quantified with Eq. (3):

$$\% \text{ removal} = 1 - \frac{[Nd \text{ or } Hf]_{measured}}{[Nd \text{ or } Hf]_{conservative}} \times 100 \tag{3}$$

where $[Nd \text{ or } Hf]_{measured}$ represents measured concentrations and $[Nd \text{ or } Hf]_{conservative}$ represents concentrations expected from two-endmember conservative mixing for the same salinity. The three-endmember mixing model is established based on the $^{143}Nd/^{144}Nd$ and $^{176}Hf/^{177}Hf$ ratios with corresponding [Nd] and [Hf] and their salinities of three dissolved sources (i.e., Amazon River, Pará River and Atlantic seawater), which are list in Table 1. Three major dissolved Nd fractions are defined: the Amazon River freshwater fraction, $f_{Ama}$; the fraction from the Atlantic seawater endmember, $f_{Atl}$; and the Pará River freshwater fraction, $f_{Pará}$. The fractions of $f_{Ama}, f_{Atl}, f_{Pará}$ of all samples along the estuarine gradient are listed in Supplementary Data 1 and are calculated by the Eqs. (4), (5), (6) and (7):

$$f_{Ama} + f_{Pará} + f_{Atl} = 1 \tag{4}$$

$$Sal_{Ama} \times f_{Ama} + Sal_{Pará} \times f_{Pará} + Sal_{Atl} \times f_{Atl} = Sal_{sample} \tag{5}$$

The Pará riverine Nd and Hf fractions are defined by Eq. (8):

$$\text{Pará riverine Nd and Hf fraction} = \frac{f_{Pará}}{(f_{Pará} + f_{Ama})} \tag{8}$$

### Discharge-weighted mean river dissolved Nd concentration and removal percentage calculations

The discharge-weighted mean dissolved [Nd] (C) of rivers was calculated by Eq. (9):

$$C = \frac{\sum_{k=1}^{n} D_k N_k}{\sum_{k=1}^{n} D_k} \tag{9}$$

where $D_k$ is the fraction of each river discharge in the total global river discharge of $133.1 \times 10^4 \, m^3 \, s^{-1}$ and $N_k$ is the [Nd] measured in each river. The discharge-weighted mean maximum estuarine Nd removal percentage (F) was calculated using Eq. (10):

$$F = \frac{\sum_{k=1}^{n} D_k R_k}{\sum_{k=1}^{n} D_k} \tag{10}$$

where $D_k$ is as above and $R_k$ is the Nd removal percentage calculated for each river.

## Data availability

All data generated in this study are provided in the Supplementary files and are also available on PANGAEA. Global river pH and dissolved organic carbon datasets (GEMStat and GLORICH) used in this study are

available through the GEMStat website (https://gemstat.org/) and PANGAEA website, respectively.

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

## Acknowledgements

We thank Andrea Koschinsky, the captain and crew of RV Meteor for their help and support during cruise M147. We also thank Jutta Heinze, Sieglinde Kolbrink, Marcus Gutjahr and Christopher Siebert for laboratory support and Lisa Bretschneider for sampling as well as Te Liu and Jörg Rickli for helpful discussions. We thank Carlos Eduardo de Rezende for the suspended matter concentration data. We also acknowledge Tristan C.C. Rousseau and Jeroen E. Sonke for their valuable advice on the mixing calculations. The China Scholarship Council (CSC) is acknowledged for financial support of A.X. during this study. G.L. gratefully acknowledges financial support from the Ocean Frontier Institute under a Canada First Research Excellence Fund award.

## Author contributions

A.X., M.F., and E.H. designed and coordinated the study. M.F. and E.H. conducted the sampling. A.X. carried out the analytical work with guidance from E.H. and G.L. A.X. wrote the manuscript. All coauthors contributed to the final version.

## Funding

## Competing interests

The authors declare no competing interests.
