## [Peer Review File · Nature Communications]

Overlooked riverine contributions of dissolved neodymium and hafnium to the Amazon estuary and oceansREVIEWER COMMENTS

Reviewer #1 (Remarks to the Author):

Review of Overlooked riverine contributions of dissolved neodymium and hafnium to the Amazon estuary and the oceans

Comments to the Editor and authors:

The present manuscript describes investigations to evaluate Nd and Hf sources and behavior from the Amazon hydrological basin to the Atlantic Ocean.

The topic is very interesting and lies within the scope of Nature communications. The introduction and state-of-the-art are excellent.

The manuscript is written with good grammar and syntax. All tables and figures are useful and self-understanding, however some little improvements are possible (see specific comments).

The methodology used agrees with higher procedure standards for traces and isotope measurement.

The global quality of data presented is high. The dataset supports the interpretations and conclusions.

This study is very important to improve our knowledge of the global cycle of trace elements, especially the contribution of the Amazon basin to Nd and Hf oceanic budget. The paper's contribution is high and will significantly contribute to the global trace elements budget issue from continents to oceans. The study is highly interesting and innovative, with important data (trace elements Isotopes) for understanding trace element behavior in the critical zone. The paper will make a valuable contribution.

The manuscript is suitable for publication in its present form.

Specific comments:

Are the authors explored geology and/or soil characteristics to explain the large Nd and Hf contribution from the Para River?

For example, L 180-181. The increase in dissolved [Nd] and [Hf] at mid to high salinities accompanied by highly unradiogenic ϵNd and ϵHf values can help to indicate the Nd and Hf origin? Can be related to the geologic age of the SPM origin?

Regarding the significant contribution of Fe and Mn hydroxides dissolution, did authors consider the modification of the isotopic composition of SPM from the source to the ocean? Based on Nd isotope composition, authors have shown that the main contribution of Mn and Fe hydroxides was the Andean tributaries. However, previous studies have shown the significant contribution of floodplain functioning, with early diagenesis processes, on the Fe and Mn cycling. Can the Nd and Hf isotopic composition associated with Fe and Mn hydroxides be altered by oxy-reduction processes occurring in the floodplain? Can the annual floodplain biogeochemical cycling explain part of the Nd and Hf temporal variation in the estuary and ocean waters? Especially Hf isotopes, as fractionations can occur during biogeochemical reactions.

Barroux G, et al. Seasonal dissolved rare earth element dynamics of the Amazon River main stem, its tributaries, and the Curuaí floodplain. *Geochemistry, Geophysics, Geosystems* 7, Q12005 (2006).

- Table 1 and figure 1 and 4: author can add a salinity unit

The water discharge value is a mean annual value? Can authors add a reference?

- Figure 2 Authors can improve the colored salinity scale. For example, the Orange color of several REE patterns is not clearly distinguishable on the salinity scale below.

The Nd and Hf removal seems mainly occur at the early salinity increase (between 1 and 3 ppt). Can it help to trace the global Nd and Hf removal process?

What is the salinity of the BATS 15m water sample?

- Fig 5d, some values are not clear regarding the pie chart icons. Can authors check the numeric value, notably some '100%'
- Figure 6c: the Pará and Amazon points should be red and black, respectively.

Reviewer #2 (Remarks to the Author):

Xu et al., present evidence that the Para River has a significant impact on Nd and Hf concentrations and isotope ratios on the Amazon Shelf. This is interesting and important because previous work has suggested that the Nd distribution on this shelf was being affected by particle dissolution. In fact, globally shelf sediment inputs of Nd (from particle dissolution or submarine groundwater discharge [SGD]) are thought to be an important part of delivery of Nd to the ocean. Xu et al. demonstrate that the Para River (which contributes about 10% of the fresh water to this estuarine region) should not be overlooked (as it has been in the past) and that its composition provides a reasonable explanation for Nd anomalies previously attributed to shelf sediment inputs. The paper is well-written and the new data on the Para are a welcome addition to the understanding of this globally important river/estuary system.

All that said, I do have some quibbles with this manuscript. It's disappointing that all of the samples seem to be near surface samples. Since this is a two-layer system, data from near bottom waters (and perhaps even from pore waters) would be quite instructive to the overall discussion of Nd sources. Also, I found the high Para fractions offshore of the Amazon mouth to be surprising (see Fig 5; Para fractions much higher than the discharge ratio). Is there other evidence to support this...for instance satellite images or oxygen isotopes in the water? That would make the argument for high Para influence more compelling.

It would also be useful to make a back of the envelope calculation (based on water amounts and water residence times) as to what sort of flux of Nd one would need from the sediments to support the non-conservative Nd they observe....then compare that to published reports (globally) of Nd fluxes (including SGD) from shallow sediments. If the possible sediment flux is too low to have an impact, then that helps support the Para influence. But, maybe sediments are still an important component: in other words, just because the Para is important, doesn't mean that it's the only factor.

After the Para discussion, the authors then turn to the global river/estuarine flux of Nd. Oddly, there are a couple of important references they seem to neglect. Gaillardet et al. (Treatise of Geochem, 2014) estimated a global average river Nd of 1056 pM, which is much closer to the value Xu et al. propose than the other references they cite. In fact, Gaillardet et al. come up with a fluvial dissolved Nd flux that is exactly the same as Xu et al. Also, missing is mention of Adebayo et al. (FMARS 2018, <https://doi.org/10.3389/fmars.2018.00166>) who noted the comparatively low Nd removal in the Mississippi River plume and speculate on similar controlling factors as Xu et al. mention.

Finally, I note that there is considerable error in the Nd flux estimates of Xu et al. It is good that they acknowledge this, but with such large error bars on the river flux one wonders if their flux estimates truly are significantly higher than previous estimates.

Response to Reviewer comments

We thank the two Reviewers for their detailed comments. We have responded to these below in blue. Transcriptions from the revised manuscript are in *blue italic*. To refer to the individual reviewer responses, we have numbered the reviewer comments (R1_1, for Reviewer 1, comment 1, and so on).

Reviewer #1 (Remarks to the Author):

Overlooked riverine contributions of dissolved neodymium and hafnium to the Amazon estuary and the oceans Xu et al.

The present manuscript describes investigations to evaluate Nd and Hf sources and behavior from the Amazon hydrological basin to the Atlantic Ocean.

The topic is very interesting and lies within the scope of Nature communications. The introduction and state-of-the-art are excellent. The manuscript is written with good grammar and syntax. All tables and figures are useful and self-understanding, however some little improvements are possible (see specific comments). The methodology used agrees with higher procedure standards for traces and isotope measurement. The global quality of data presented is high. The dataset supports the interpretations and conclusions.

This study is very important to improve our knowledge of the global cycle of trace elements, especially the contribution of the Amazon basin to Nd and Hf oceanic budget. The paper's contribution is high and will significantly contribute to the global trace elements budget issue from continents to oceans. The study is highly interesting and innovative, with important data (trace elements Isotopes) for understanding trace element behavior in the critical zone. The paper will make a valuable contribution.

The manuscript is suitable for publication in its present form.

We sincerely thank Reviewer #1 for providing valuable feedback and dedicating time to review our manuscript. The constructive input has greatly improved the quality of our work. In the following sections, we address each of the comments and suggestions.

Specific comments and suggestions are listed below identified by numbers

R1_1:

Are the authors explored geology and/or soil characteristics to explain the large Nd and Hf contribution from the Para River?

For example, L 180-181. The increase in dissolved [Nd] and [Hf] at mid to high salinities accompanied by highly unradiogenic ϵ_{Nd} and ϵ_{Hf} values can help to indicate the Nd and Hf origin? Can be related to the geologic age of the SPM origin?

The relatively unradiogenic ϵ_{Nd} and ϵ_{Hf} values at mid to high salinities strongly suggest that the Pará River, which is less affected by contributions from the Andes and consequently exhibits unradiogenic ϵ_{Nd} and ϵ_{Hf} signatures of -14.1 ± 0.2 and -4.1 ± 0.6 , respectively, is the main source of the estuarine signatures. Beyond that, it is difficult to relate these unradiogenic ϵ_{Nd} and ϵ_{Hf} values of the estuarine waters to the signatures of parent rocks in the drainage basin due to data scarcity upstream of the Pará River and because the ϵ_{Nd} and ϵ_{Hf} signatures of the riverine dissolved loads are usually more radiogenic than the suspended sediment loads or their source rocks due to preferential alteration of marine precipitates (Goldstein and Jacobsen, 1987; Bayon et al., 2006, 2020), incongruent weathering (Aubert et al., 2001; Viers and Wasserburg, 2004; Bayon et al., 2015) and the zircon effect for Hf isotopes (Patchett et al., 1984). Solely based on the contrast in signatures between the Amazon and Pará rivers, however, we can infer that the origin of the less radiogenic ϵ_{Nd} and ϵ_{Hf} signatures of the Pará River is the cratonic Shield, whose parent rock and suspended particulate matter (SPM) ϵ_{Nd} signatures range from -16 to -24 (Merschel et al., 2017a; Höppner et al., 2018; Horbe et al., 2022). The high [Nd] and [Hf] are likely not a function of the source rock compositions but are mainly caused by the lower pH and high [Nd] and [Hf] concentrations in the tributaries from the mangrove forest area, which is consistent with high trace metal exports from Amazonian mangrove forest area (de Carvalho et al., 2021; Hollister et al., 2022; Matos et al., 2022).

We would like to stress that the lack of precise knowledge regarding the specific contributions of hinterland endmembers to the dissolved composition of the Pará River does not impact the core finding of our study, which is that the predominant factor controlling the radiogenic isotope composition of the dissolved load in the Amazon estuary is the mixing of water masses. However, following the reviewer's comment, we added some discussion on the potential drivers of the large Nd and Hf contribution from the Pará River as follows:

Line 85-88: In addition, parent rock characteristics and floodplain supply may play a role given that elevated REY concentrations in the waters exiting the floodplain have been observed in the Amazon Basin³¹.

Line 128-133: The dissolved ϵ_{Nd} and ϵ_{Hf} signatures of Amazon River freshwater are -9.4 ± 0.2 and $+1.8 \pm 0.9$, respectively, while those of Pará River freshwater are markedly less radiogenic, reaching -14.1 ± 0.2 and -4.1 ± 0.6 , respectively. These Pará River signatures are likely associated with higher contributions from weathering of the cratonic Shield, whose parent rocks and suspended particulate matter (SPM) ϵ_{Nd} signals predominantly range from -16 to -24 (refs 41, 42, 43).

References:

- Aubert D, Stille P, Probst A. REE fractionation during granite weathering and removal by waters and suspended loads: Sr and Nd isotopic evidence. *Geochim Cosmochim Acta* 65, 387-406 (2001).
- Bayon G, et al. Rare earth element and neodymium isotope tracing of sedimentary rock weathering. *Chem Geol*, 119794 (2020).
- Bayon G, et al. Rare earth elements and neodymium isotopes in world river sediments revisited. *Geochim Cosmochim Acta* 170, 17-38 (2015).
- Bayon G, Vigier N, Burton KW, Jean Carignan AB, Etoubleau J, Chu N-C. The control of weathering processes on riverine and seawater hafnium isotope ratios. *Geology* 34, 433-436 (2006).
- de Carvalho LM, Hollister AP, Trindade C, Gledhill M, Koschinsky A. Distribution and size fractionation of nickel and cobalt species along the Amazon estuary and mixing plume. *Marine Chemistry* 236, 104019 (2021).
- Goldstein SJ, Jacobsen SB. The Nd and Sr isotopic systematics of river-water dissolved material: Implications for the sources of Nd and Sr in seawater. *Chemical Geology: Isotope Geoscience section* 66, 245-272 (1987).
- Hollister AP, Leon M, Scholten J, Van Beek P, Gledhill M, Koschinsky A. Distribution and Flux of Trace Metals in the Amazon and Pará River Estuary and Mixing Plume. Authorea, <https://doi.org/10.1002/essoar.10512637.10512631> (2022).
- Höppner N, Lucassen F, Chiessi CM, Sawakuchi AO, Kasemann SA. Holocene provenance shift of suspended particulate matter in the Amazon River basin. *Quat Sci Rev* 190, 66-80 (2018).
- Horbe AMC, Albuquerque MFDS, Dantas EL. Nd and Sr Isotopes and REE Investigation in Tropical Weathering Profiles of Amazon Region. *Frontiers in Earth Science* 10, (2022).
- Matos CRL, et al. Seasonal changes in metal and nutrient fluxes across the sediment-water interface in tropical mangrove creeks in the Amazon region. *Applied Geochemistry* 138, 105217 (2022).
- Merschel G, Bau M, Schmidt K, Münker C, Dantas EL. Hafnium and neodymium isotopes and REY distribution in the truly dissolved, nanoparticulate/colloidal and suspended loads of rivers in the Amazon Basin, Brazil. *Geochim Cosmochim Acta* 213, 383-399 (2017a).

- Patchett P, White W, Feldmann H, Kielinczuk S, Hofmann A. Hafnium/rare earth element fractionation in the sedimentary system and crustal recycling into the Earth's mantle. *Earth Planet Sci Lett* 69, 365-378 (1984).
- Viers J, Wasserburg GJ. Behavior of Sm and Nd in a lateritic soil profile. *Geochim Cosmochim Acta* 68, 2043-2054 (2004).

R1_2:

Regarding the significant contribution of Fe and Mn hydroxides dissolution, did authors consider the modification of the isotopic composition of SPM from the source to the ocean? Based on Nd isotope composition, authors have shown that the main contribution of Mn and Fe hydroxides was the Andean tributaries. However, previous studies have shown the significant contribution of floodplain functioning, with early diagenesis processes, on the Fe and Mn cycling. Can the Nd and Hf isotopic composition associated with Fe and Mn hydroxides be altered by oxy-reduction processes occurring in the floodplain?

We agree with the reviewer that early diagenetic processes in the floodplain influence the Fe-Mn cycling, potentially increasing the concentrations of dissolved REY in the waters exiting the floodplain (e.g., Barroux et al., 2006). However, these processes would not change the radiogenic Nd isotopes and mainly impact the dissolved load due to the dissolution of Fe-Mn oxyhydroxides and/or potential particle-water interaction. The ϵ_{Nd} signatures of the Fe-Mn oxyhydroxide phase of SPM range from -8.4 to -8.1 in this study, which are intermediate between the ϵ_{Nd} signatures of the Solimões (-7.1) and Madeira (-10.0) tributaries (Merschel et al., 2017a), suggesting that the Fe-Mn oxyhydroxide fraction mainly reflects the isotopic signals of these Andean tributaries thus requiring no further modification through floodplain processes. This is also supported by the fact that 84% of the total amount of dissolved salts and suspended solids discharged via the Amazon River are eroded from the Andean area (Gibbs, 1967). Thus, based on our data, we infer that the ϵ_{Nd} and ϵ_{Hf} signatures of the Fe-Mn oxyhydroxide phase of SPM in the Amazon River are not significantly affected by the oxy-reduction processes in the floodplain.

References:

- Barroux G, et al. Seasonal dissolved rare earth element dynamics of the Amazon River main stem, its tributaries, and the Curuaí floodplain. *Geochemistry, Geophysics, Geosystems* 7, Q12005 (2006).
- Gibbs RJ. The Geochemistry of the Amazon River System: Part I. The Factors that Control the Salinity and the Composition and Concentration of the Suspended Solids. *Geol Soc Am Bull* 78, 1203 (1967).

Merschel G, Bau M, Schmidt K, Münker C, Dantas EL. Hafnium and neodymium isotopes and REY distribution in the truly dissolved, nanoparticulate/colloidal and suspended loads of rivers in the Amazon Basin, Brazil. *Geochim Cosmochim Acta* 213, 383-399 (2017a).

R1_3:

Can the annual floodplain biogeochemical cycling explain part of the Nd and Hf temporal variation in the estuary and ocean waters? Especially Hf isotopes, as fractionations can occur during biogeochemical reactions.

We agree that seasonal variability in floodplain biogeochemical cycling influences the Fe-Mn cycling and can increase the dissolved REY concentrations, as also demonstrated by Barroux et al., 2006. This may have implications for dissolved REY and Hf concentrations of Amazon freshwater, which we now mention in the manuscript (see below). However, the effects of floodplain processes on the Nd isotope compositions are difficult to verify since the ϵ_{Nd} value of the Amazon freshwater endmember in our study (-9.4 ± 0.2) agrees well with previously measured values as given in the manuscript, which are -9.2 ± 0.4 (Stordal and Wasserburg, 1986), -8.9 ± 0.5 (Piepgras and Wasserburg, 1987) and -8.8 ± 0.2 (Rousseau et al., 2015). For Hf isotopes, there are no data available from previous studies in the Amazon estuary for comparison. The ϵ_{Hf} of Amazon freshwater ($+1.8 \pm 0.9$) in our study is indeed less radiogenic than that of the main upstream tributary (Solimões, $+6.4$) (Merschel et al., 2017a), which may be related to a contribution from downstream tributaries, exchange between river water and secondary minerals and/or influence of biogeochemical changes in the floodplain. Due to the scarcity of data, we are, however, unable to demonstrate that the annual variability floodplain biogeochemical cycling has a significant impact on the ϵ_{Hf} signatures in the estuary and thus refrain from any further speculations.

Line 88-92: The dissolved [Nd] of Amazon River water sampled in April-May 2018 agrees well with the values of 471-579 pmol kg⁻¹ reported for August 1989 (ref. 32) but is substantially lower than the 850 pmol kg⁻¹ found in April 2008 (ref. 13) documenting a dynamic mixing regime in the estuary and significant interannual variability, which may be related to biogeochemical changes in the floodplain³¹.

References:

- Barroux G, et al. Seasonal dissolved rare earth element dynamics of the Amazon River main stem, its tributaries, and the Curuaí floodplain. *Geochemistry, Geophysics, Geosystems* 7, Q12005 (2006).
- Merschel G, Bau M, Schmidt K, Münker C, Dantas EL. Hafnium and neodymium isotopes and REY distribution in the truly dissolved, nanoparticulate/colloidal and suspended loads of rivers in the Amazon Basin, Brazil. *Geochim Cosmochim Acta* 213, 383-399 (2017a).
- Piegras D, Wasserburg G. Rare earth element transport in the western North Atlantic inferred from Nd isotopic observations. *Geochim Cosmochim Acta* 51, 1257-1271 (1987).
- Rousseau TC, et al. Rapid neodymium release to marine waters from lithogenic sediments in the Amazon estuary. *Nat Commun* 6, 7592 (2015).
- Stordal M, Wasserburg G. Neodymium isotopic study of Baffin Bay water: sources of REE from very old terranes. *Earth Planet Sci Lett* 77, 259-272 (1986).

R1_4:

Table 1 and figure 1 and 4: author can add a salinity unit

Although salinity is generally given without units, we have added the unit “psu” in tables and figures in the revised manuscript.

R1_5:

The water discharge value is a mean annual value? Can authors add a reference?

Yes, the water discharge value represents the mean annual value and corresponding references have been added in the revised manuscript:

Line 695-696 in figure caption: The mean annual freshwater discharges of the Amazon River and Pará River are indicated^{10, 26}.

R1_6:

Figure 2 Authors can improve the colored salinity scale. For example, the Orange color of several REE patterns is not clearly distinguishable on the salinity scale below.

We changed the salinity color scale to make the REE patterns better distinguishable.

R1_7:

The Nd and Hf removal seems mainly occur at the early salinity increase (between 1 and 3 ppt). Can it help to trace the global Nd and Hf removal process?

The large-scale removal of dissolved Nd and Hf generally is well-known to occur in the low-salinity zone of riverine estuaries but depends on the content of inorganic

(nano-)particles and colloids (NPCs) and dissolved organic carbon (DOC) (Hoyle et al., 1984; Goldstein and Jacobsen, 1988; Sholkovitz, 1993). In estuaries of rivers with high DOC and low NPCs concentrations, the Nd and Hf can display conservative mixing with seawater (Merschel et al., 2017b; Rahlf et al., 2021). Therefore, the removal process in the Amazon estuary can only be considered an example of high NPC and low DOC influence and is unsuitable for tracing the Nd and Hf removal processes in all estuaries. In section “Implications for global riverine dissolved Nd and Hf fluxes” we provide a new estimate (which agrees very well with previous averages) for the global discharge weighted removal of both elements.

References:

- Goldstein SJ, Jacobsen SB. REE in the Great Whale River estuary, northwest Quebec. *Earth Planet Sci Lett* 88, 241-252 (1988).
- Hoyle J, Elderfield H, Gledhill A, Greaves M. The behaviour of the rare earth elements during mixing of river and sea waters. *Geochim Cosmochim Acta* 48, 143-149 (1984).
- Merschel G, Bau M, Dantas EL. Contrasting impact of organic and inorganic nanoparticles and colloids on the behavior of particle-reactive elements in tropical estuaries: An experimental study. *Geochim Cosmochim Acta* 197, 1-13 (2017).
- Rahlf P, Laukert G, Hathorne EC, Vieira LH, Frank M. Dissolved neodymium and hafnium isotopes and rare earth elements in the Congo River Plume: Tracing and quantifying continental inputs into the southeast Atlantic. *Geochim Cosmochim Acta* 294, 192-214 (2021).
- Sholkovitz ER. The geochemistry of rare earth elements in the Amazon River estuary. *Geochim Cosmochim Acta* 57, 2181-2190 (1993).

R1_8:

What is the salinity of the BATS 15m water sample?

The salinity of BATS 15m water is 36.5, which has now been added in the revised manuscript.

R1_9:

Fig 5d, some values are not clear regarding the pie chart icons. Can authors check the numeric value, notably some ‘100%’

Thank you for noticing this error. Fig. 5 has been revised and this issue has been clarified. The percentage numbers in Fig. 5c and Fig. 5d are Pará riverine Nd fraction and Hf fraction, defined as $f_{\text{Pará}}/(f_{\text{Pará}}+f_{\text{Ama}})$, to compare the relative contributions of isotopes from the Amazon River and Pará River. Thus, if f_{Ama} is close to 0, the

percentage will be 100% and suggest that Pará River water mixed with seawater dominates the isotopic signatures of the respective estuarine water sample.

We added related discussions for better understanding:

Line 204-208: To compare the relative contribution of isotope signatures from the Amazon and Pará rivers, the riverine Nd and Hf proportion originating from the Pará River (named Pará riverine Nd fraction and Hf fraction) in each water sample was calculated and displayed numerically (Fig. 5c and 5d).

Line 736-739: in the figure caption: In panels c) and d), fractions of Pará River water, Amazon River water and Atlantic seawater in the Amazon estuary are represented as pie charts calculated by equations (2), (3), and (4). The Pará riverine Nd and Hf fractions are defined by equation (5) in Methods and displayed numerically.

Line 423-424 in Methods: The Pará riverine Nd and Hf fractions are defined by equation (5):

$$\text{Pará riverine Nd and Hf fraction} = f_{\text{Pará}} / (f_{\text{Pará}} + f_{\text{Ama}}) \quad (5)$$

R1_10:

Figure 6c: the Pará and Amazon points should be red and black, respectively.

We marked these points accordingly. See revised Fig. 6.

Reviewer #2 (Remarks to the Author):

Xu et al., present evidence that the Para River has a significant impact on Nd and Hf concentrations and isotope ratios on the Amazon Shelf. This is interesting and important because previous work has suggested that the Nd distribution on this shelf was being affected by particle dissolution. In fact, globally shelf sediment inputs of Nd (from particle dissolution or submarine groundwater discharge [SGD]) are thought to be an important part of delivery of Nd to the ocean. Xu et al. demonstrate that the Para River (which contributes about 10% of the fresh water to this estuarine region) should not be overlooked (as it has been in the past) and that its composition provides a reasonable explanation for Nd anomalies previously attributed to shelf sediment inputs. The paper is well-written and the new data on the Para are a welcome addition to the understanding of this globally important river/estuary system.

We thank Reviewer #2 for her/his positive comments and recommendations below that have led to a significant improvement of the manuscript. We have implemented several changes in the revised version in response to these recommendations, including reporting the Nd and Hf isotopes of seven near-bottom water samples to constrain a potential bottom source, setting up a box model to clarify the contribution of the Pará River to the Amazon estuary and revising the last section of the manuscript. Detailed responses to each comment are provided below.

R2_1:

It's disappointing that all of the samples seem to be near surface samples. Since this is a two-layer system, data from near bottom waters (and perhaps even from pore waters) would be quite instructive to the overall discussion of Nd sources.

The Reviewer makes an important point here, but all the literature data available for comparison is only from the surface layer and there is a very strong near-surface physical stratification that essentially prevents vertical exchange between the fresh surface waters and the underlying waters that are characterized by open ocean salinities (Pailler et al., 1999; Tyaquiçã et al., 2017; Varona et al., 2019). Nevertheless, following the reviewer's comment, we have analyzed Nd and Hf isotope compositions of seven near-bottom water samples (Supplementary Fig. 3), which are on average -11.0 ± 1.2 and 1.0 ± 1.7 , respectively. These averages are used in the revised manuscript as further evidence for limited exchange between the surface and bottom layers and, thus, for the admixture of the Pará River to the surface layer as the main reason for surface signature changes. The Pará River has the least radiogenic ϵ_{Nd} and ϵ_{Hf} signatures in our study (-14.1 ± 0.2 and -4.1 ± 0.6 , respectively) and, in agreement with other evidence (e.g., satellite images, see below), can be considered the main source of similarly unradiogenic Nd isotope signatures (-13.7) in the mid- to high-salinity zone. The detailed analysis of deeper water samples of the estuary will be the subject of a separate article, including a comprehensive discussion of the effects of particle dissolution and shelf exchange. We have now clarified that the focus of this study is on the processes occurring in the surface plume and the dissolved fluxes to the surface Atlantic.

We added related discussions and a figure to the revised manuscript and the supplement:

Line 180-185: To further constrain potential sedimentary Nd and Hf sources, seven near-bottom water samples recovered in the continental shelf area of the Amazon estuary (Supplementary Fig. 3) were measured. Their mean ϵ_{Nd} and ϵ_{Hf} signatures of

are $-11.0 \pm 1.2 (\pm 1s.d., n=7)$ and $1.0 \pm 1.7 (\pm 1s.d., n=7)$, respectively, excluding bottom supply as a significant source of the observed unradiogenic surface water ϵ_{Nd} and ϵ_{Hf} signatures.

Line 185-187: Particle dissolution/particle-seawater interaction (i.e., boundary exchange processes) may still occur but will be restricted to the bottom layer below the freshwater plume on the continental shelf and deep-sea fan.

Supplementary Fig. 3. Stations of near-bottom water sampling above the continental shelf area of the Amazon estuary. Salinity is given in psu.

References:

- Pailler K, Bourlès B, Gouriou Y. The barrier layer in the western tropical Atlantic Ocean. *Geophys Res Lett* 26, 2069-2072 (1999).
- Tyaquicã P, et al. Amazon Plume Salinity Response to Ocean Teleconnections. *Frontiers in Marine Science* 4, 250 (2017).
- Varona HL, Veleda D, Silva M, Cintra M, Araujo M. Amazon River plume influence on Western Tropical Atlantic dynamic variability. *Dynamics of Atmospheres and Oceans* 85, 1-15 (2019).

R2_2:

Also, I found the high Para fractions offshore of the Amazon mouth to be surprising (see Fig 5; Para fractions much higher than the discharge ratio). Is there other evidence to support this...for instance satellite images or oxygen isotopes in the water? That would make the argument for high Para influence more compelling.

The high Pará fractions in the outer estuary are consistent with the measured river water concentrations and the expected pattern of regional coastal circulation (Barnier et al., 2001; Prestes et al., 2018). The Amazon plume is usually deflected northward along the coast, while the direct deflection of the Pará plume is hindered by the strong discharge of the Amazon River. However, further offshore, the highly channeled Pará plume is also deflected to the north by the vigorous North Brazil Current (NBC), which explains the observed higher fractions of the Pará River at this location. This Pará plume propagation can also be inferred from satellite images of mud distribution (Figure below). But detection of this pattern using oxygen isotopes is difficult, as end-member values of both rivers are too similar to allow distinction of the riverine contributions in the estuary (Karr and Showers, 2002). The importance of the Pará River for trace metal supply to the Amazon plume is further supported by data for other metals from the same cruise (e.g., Ni, Co, Ti, Al, Zn, Pb) (de Carvalho et al., 2021; Hollister et al., 2022; Schneider et al., 2022).

We added the following discussion to the revised manuscript:

Line 214-220: The large and previously overlooked dissolved Nd and Hf contributions from the Pará River are supported by the regional coastal circulation and satellite images of mud distribution in the estuary (Supplementary Fig. 6), as well as high concentrations of other trace metals (Fe, Ni, Co, Ti, Al, Zn, Pb) in the Pará River^{27, 28, 29}. Therefore, the Pará River is an essential source of micronutrients to the Amazon estuary and to the western Atlantic and thus needs to be considered in future studies on the budget of trace elements of the western Atlantic Ocean.

Supplementary Fig. 6. Regional coastal circulation¹ and satellite images of mud distribution in the Amazon estuary², documenting the northward deflection of both river plumes. The Amazon plume is usually deflected northward along the coast, while the direct deflection along the coast of the Pará plume is hindered by the strong discharge of the Amazon River. However, further offshore, the highly channeled Pará plume is also deflected to the north by the vigorous North Brazil Current (NBC), which explains the observed higher Pará River fractions at this location.

References:

Barnier B, et al. On the seasonal variability and eddies in the North Brazil Current: insights from model intercomparison experiments. *Progress in Oceanography* 48, 195-230 (2001).

- de Carvalho LM, Hollister AP, Trindade C, Gledhill M, Koschinsky A. Distribution and size fractionation of nickel and cobalt species along the Amazon estuary and mixing plume. *Marine Chemistry* 236, 104019 (2021).
- Hollister AP, Leon M, Scholten J, Van Beek P, Gledhill M, Koschinsky A. Distribution and Flux of Trace Metals in the Amazon and Pará River Estuary and Mixing Plume. Authorea, <https://doi.org/10.1002/essoar.10512637.10512631> (2022).
- Karr JD, Showers WJ. Stable oxygen and hydrogen isotopic tracers in Amazon shelf waters during Amassed. *Oceanologica acta* 25, 71-78 (2002).
- Prestes YO, Silva ACd, Jeandel C. Amazon water lenses and the influence of the North Brazil Current on the continental shelf. *Continental Shelf Research* 160, 36-48 (2018).
- Schneider AB, Koschinsky A, Krause CH, Gledhill M, de Carvalho LM. Dynamic behavior of dissolved and soluble titanium along the salinity gradients in the Pará and Amazon estuarine system and associated plume. *Marine Chemistry* 238, 104067 (2022).

R2_3:

It would also be useful to make a back of the envelope calculation (based on water amounts and water residence times) as to what sort of flux of Nd one would need from the sediments to support the non-conservative Nd they observe....then compare that to published reports (globally) of Nd fluxes (including SGD) from shallow sediments. If the possible sediment flux is too low to have an impact, then that helps support the Para influence. But, maybe sediments are still an important component: in other words, just because the Para is important, doesn't mean that it's the only factor.

The near-bottom water and suspended particle isotopic data we have added provide strong evidence to exclude the influence of particles/sediments. However, following the reviewer's comment, we have established a box model of the Amazon estuary (Supplementary Fig. 4 and 5) to clarify the contribution of the Pará River in the Amazon estuary and to constrain the key role of the Pará River in supplying unradiogenic Nd and Hf isotopes to the southern and outer Amazon estuary. The result demonstrates that admixture of Pará River water can shift the ϵ_{Nd} and ϵ_{Hf} values in the outer Amazon estuary to values of -13.9 ~ -13.7 and -4.1 ~ -3.6, respectively, which are identical within error to the measured values without invoking any additional more radiogenic sedimentary source (i.e., near-bottom waters and suspended particles).

We have now added these findings and figures to the revised manuscript and the Supplementary Information:

Line 187-196: Based on the above evidence, admixture of the Pará River water, which has the highest dissolved [Nd] and [Hf] (1036 pmol kg⁻¹ and 13.4 pmol kg⁻¹, respectively) and least radiogenic ϵ_{Nd} and ϵ_{Hf} signatures (-14.1 ± 0.2 , -4.1 ± 0.6 , respectively, Fig. 4) is the most likely explanation for the shift in isotopic signatures to highly unradiogenic values along the salinity gradient of the Amazon surface water plume. This is supported by a box model (Supplementary Figs. 4 and 5), showing that admixture of Pará River water can indeed shift the ϵ_{Nd} and ϵ_{Hf} in the outer Amazon estuary to values of $-13.9 \sim -13.7$ and $-4.1 \sim -3.6$, respectively, which are identical within error to the measured values, indicating an additional sedimentary source is not required to explain the data.

Line 87-103 in the Supplementary Information:

Establishment of the box model

The box model was established following Kaul and Froelich⁶. The relationships between salinity and dissolved Nd and Hf concentrations with distance in the estuary are described by the following equations:

$$F(Nd) = f(x) = 318.056 \times x^{-0.442} \quad (1)$$

$$F(Hf) = f(x) = 7.455 \times x^{-0.394} \quad (2)$$

$$F(\text{Salinity}) = f(x) = 1.047 \times x^{0.549} \quad (3)$$

The freshwater volume, seawater volume, Nd and Hf masses in the Pará estuary and outer Amazon estuary were calculated by integrating the following equations:

$$\text{Nd or Hf mass } (M_{Nd \text{ or Hf-final}}): \int_{0.1}^{\text{distance}} f(x) \cdot \text{depth} \cdot \text{width} \cdot dx \quad (4)$$

$$\text{Seawater volume } (Vol_{\text{seawater}}): \int_{0.1}^{\text{distance}} \frac{(f(x) - Sal_{\text{freshwater}})}{Sal_{\text{seawater}}} \text{depth} \cdot \text{width} \cdot dx \quad (5)$$

Freshwater volume ($Vol_{\text{freshwater}}$):

$$\int_{0.1}^{\text{distance}} \frac{\{Sal_{\text{seawater}} - f(x)\}}{Sal_{\text{seawater}} - Sal_{\text{freshwater}}} \text{depth} \cdot \text{width} \cdot dx \quad (6)$$

ϵ_{Nd} or ϵ_{Hf} was calculated by equation (7):

$$\epsilon_{Nd_{\text{final}}} = \frac{M_{Nd-\text{freshwater}} \times \epsilon_{Nd_{\text{freshwater}}} + M_{Nd-\text{seawater}} \times \epsilon_{Nd_{\text{seawater}}}}{M_{Nd-\text{final}} + M_{Nd-\text{removal}}} \quad (7)$$

where the $M_{Nd \text{ or Hf-freshwater}}$ or $-seawater$ is the gross Nd or Hf mass supplied by the Amazon River, Pará River or seawater and is calculated by equation (8), $M_{Nd \text{ or Hf-final}}$ is the Nd

or Hf mass in the box calculated by equation (4) and $M_{Nd \text{ or } Hf\text{-removal}}$ is the Nd or Hf mass removed in the estuary calculated by equation (9):

$$M_{Nd \text{ or } Hf} = Vol_{water} \times [Nd] \text{ or } [Hf] \quad (8)$$

$$M_{Nd \text{ or } Hf\text{-removal}} = Vol_{water} \times [Nd]_{freshwater} \text{ or } [Hf]_{freshwater} \times Percentage_{removal} \quad (9)$$

where the $Percentage_{removal}$ reflects the maximum removal percentage of Nd or Hf in the estuary quantified with equation (1) in the main text.

Supplementary Fig. 4. Establishment of a box model for the Amazon estuary. The box includes the Pará estuary and outer Amazon estuary with width W_B . The schematic of the Nd and Hf mass calculations in the box model is shown in panel b.

Supplementary Fig. 5. Relationships between salinity and dissolved Nd and Hf concentrations with distance from the Pará River mouth.

Supplementary Table 4. Parameters and results of the box model used in this study.

Parameters or results	Pará River	Amazon River	Seawater
	Box	Box	
[Nd], [Hf] (pmol kg^{-1})	1035.6, 13.4	501.9, 12.3	25.8, 0.4
ϵ_{Nd} , ϵ_{Hf}	-14.0, -4.1	-9.4, 1.8	-11.4, -1.0
Salinity	0.4	0.2	35.5
Distance, width and depth (km)	352, 70, 0.005	352, 70, 0.005	
Discharge in May ($\text{m}^3 \text{ day}^{-1}$)	2.6×10^9	2.3×10^{10}	
Seawater volume (m^3)	5.7×10^{10}	5.7×10^{10}	
Freshwater volume (m^3)	6.5×10^{10}	7.5×10^9	
Nd, Hf mass (mol)	5.2×10^3 , 1.5×10^2	5.2×10^3 , 1.5×10^2	
Nd, Hf maximum removal fraction (%)	95.0%, 82.5%	90.8%, 87.3%	
Gross Nd, Hf supply (mol)	6.8×10^4 , 8.7×10^2	$0-3.8 \times 10^3$, 0-93	
Removed Nd, Hf mass (mol)	6.4×10^4 , 7.2×10^2	$0-3.4 \times 10^3$, 0-81	
Net Nd, Hf supply (mol)	3.4×10^3 , 1.5×10^2	$0-3.5 \times 10^2$, 0-12	1.5×10^3 , 23
Calculated mean ϵ_{Nd} , ϵ_{Hf}	-13.9 ~ -13.7, -4.1 ~ -3.6		

R2_4:

After the Para discussion, the authors then turn to the global river/estuarine flux of Nd. Oddly, there are a couple of important references they seem to neglect. Gaillardet et al. (Treatise of Geochem, 2014) estimated a global average river Nd of 1056 pM, which is

much closer to the value Xu et al. propose than the other references they cite. In fact, Gaillardet et al. come up with a fluvial dissolved Nd flux that is exactly the same as Xu et al.

We thank the reviewer for pointing out an important publication that had not been included and which we had missed. We have now added the reference in the manuscript:

Line 259-264: Considering a global river discharge of $133.1 \times 10^4 \text{ m}^3 \text{ s}^{-1}$ (Ref. 57), a global riverine flux of $5.7 \times 10^9 \text{ g Nd yr}^{-1}$ is calculated, which is (nearly) identical to previous estimates in two studies that took the [Nd] of only 40 or 21 rivers into account (5.4 or $5.7 \times 10^9 \text{ g Nd yr}^{-1}$, respectively)^{62,63}. Using a larger global dataset ($n=582$), our study confirms the few previous estimates of the dissolved riverine [Nd] flux while increasing data coverage by over 10-fold.

R2_5:

Also, missing is mention of Adebayo et al. (FMARS 2018, <https://doi.org/10.3389/fmars.2018.00166>) who noted the comparatively low Nd removal in the Mississippi River plume and speculate on similar controlling factors as Xu et al. mention.

We regret this oversight and appreciate the help of the reviewer in pointing this out.

Line 268-272: As revealed by mixing experiments³⁸ and a lower Nd removal percentage (~50%) in the Mississippi River estuary attributed to strong aqueous complexation of REY with natural organic ligands and carbonate ions⁶⁴, we find that maximum Nd removal is closely related to dissolved organic carbon concentration ([DOC]) based on the 12 available estuarine transects.

R2_6:

Finally, I note that there is considerable error in the Nd flux estimates of Xu et al. It is good that they acknowledge this, but with such large error bars on the river flux one wonders if their flux estimates truly are significantly higher than previous estimates.

We thank the reviewer for bringing attention to the considerable uncertainties in the Nd flux estimates. It is an important point to consider. In our study, the standard deviation ($\pm 1\text{s.d.}$) represents the statistical dispersion of [Nd] in global rivers, i.e., the large 1s.d. here suggests a wide range of values in the dataset ($n=582$), as shown by the positively skewed distribution of [Nd] in global rivers (Fig. 6b). It must be acknowledged that this

estimate is still limited by the small dataset of combined [Nd] and pH values in global rivers. Therefore, we calculated the weighted mean instead of the mean value to reduce the effect of extreme values. Nevertheless, we found that it was not appropriate to solely rely on the standard deviation for displaying the accuracy and uncertainty of the weighted mean calculation. To address this concern and provide a clearer representation of the precision of our calculation, we have now included the standard error (1s.e.) and 95% confidence interval (CI) in the revised manuscript (see below). This addition allows for a more comprehensive understanding of the estimated flux and the associated uncertainty. Furthermore, it is worth noting that our calculated value of 943 pmol kg⁻¹ is consistent with the estimates of 1054 pmol kg⁻¹ by Gaillardet et al. (2003) and 894 pmol kg⁻¹ by Dang et al. (2021). These previous studies did not account for uncertainties in the same manner as our study, nor did they incorporate constraints from as many rivers. The close correspondence of our estimate to these previous values supports the reliability and accuracy of our calculation.

We acknowledge the need for improved clarity and appreciate the opportunity to address this concern. By incorporating the standard error and confidence interval, we now provide a more robust and accurate representation of the precision of our calculations. This revision strengthens our findings and ensures transparency in reporting the uncertainties associated with the estimated Nd flux.

We have included related discussions in the revised manuscript, as well as in the Supplemental Information.

Line 757-762 in table caption: Table 2. Comparison of optimized Nd parameters between the literature and our study. Previously used [Nd] observed, [Removal] observed, gross, net river Nd flux and global Nd budget are based on previous studies^{19, 21, 53, 54, 55}. The standard deviation (1s.d.), standard error (1s.e.) and 95% confidence interval (CI) of the calculations are listed below, and a detailed uncertainty assessment is included in the Supplementary Information.

	Previously used	Revised
		Weighted mean, 1s.d., 1s.e., 95% CI
[Nd]_{observed} (pmol kg⁻¹)	146, 284	997, 860, 36, [927, 1067]
[Nd]_{predicted} (pmol kg⁻¹)	—	943, 1534, 64, [818, 1067]
[Removal]_{observed} (%)	70	85, 13, 4, [78, 93]
[Removal]_{predicted} (%)	—	74, 9, 1, [73, 76]
Gross river Nd flux (g yr⁻¹)	0.9-1.7 × 10⁹	5.7 × 10⁹
Net river Nd flux (g yr⁻¹)	0.3-0.5 × 10⁹	1.5 × 10⁹
Global Nd budget (g yr⁻¹)	6-9 × 10⁹	

Line 105-115 in Supplement Information:

Quantifying uncertainty in the calculation of revised weighted mean riverine dissolved [Nd] using global datasets

In this study, the pH values of 582 global rivers and DOC data of 211 rivers entering the oceans have been compiled to predict [Nd] and maximum removal percentage. To assess the uncertainty in our calculations, we employed measures such as standard deviation (1s.d.), standard error (1s.e.), and 95% confidence interval (CI) (listed in table 2). The calculation of [Nd] yielded a 1s.d. of 1534 pmol kg⁻¹, confirming a wide range of values in the dataset (n=582), as shown by the positively skewed distribution of [Nd] in global rivers (Fig. 6b). Therefore, we calculated the weighted mean instead of the mean value to reduce the effect of extreme values. It is important to note that the standard deviation primarily represents the statistical dispersion of [Nd] in global rivers. Thus, it was not appropriate to solely rely on the standard deviation for displaying the accuracy and uncertainty of the weighted mean calculation. The small 1s.e. and narrow range of 95% CI suggest relatively precise estimations of dissolved riverine weighted mean [Nd] and maximum removal percentage, with low levels of uncertainty. Furthermore, it is worth noting that our calculated value of 943 pmol kg⁻¹ for the predicted weighted mean dissolved riverine [Nd] is consistent with the estimates of 1054 pmol kg⁻¹ by Gaillardet⁷ and 894 pmol kg⁻¹ by Dang⁸. The close correspondence of our estimate to these previously determined values supports the reliability and accuracy of our calculations.

References:

- Dang DH, Wang W, Sikma A, Chatzis A, Mucci A. The contrasting estuarine geochemistry of rare earth elements between ice-covered and ice-free conditions. *Geochim Cosmochim Acta* 317, 488-506 (2021).
- Gaillardet J, Viers J, Dupré B. Trace Elements in River Waters. In: *Treatise on Geochemistry* (eds Holland HD, Turekian KK). Pergamon (2003).